# Distinct changes in endosomal composition promote NLRP3 inflammasome activation

**Zhirong Zhang** [1,2,3,4,10] ✉, **Rossella Venditti** [5,6,10], **Li Ran**[1,2,3,4], **Zengzhen Liu**[1,2,3,4],
**Karl Vivot**[1,2,3,4], **Annette Schürmann** [7], **Juan S. Bonifacino**[8],
**Maria Antonietta De Matteis** [5,6] ✉ **& Romeo Ricci** [1,2,3,4,9] ✉

Inflammasome complexes are pivotal in the innate immune response. The NLR family pyrin domain containing protein 3 (NLRP3) inflammasome is activated in response to a broad variety of cellular stressors. However, a primary and converging sensing mechanism by the NLRP3 receptor initiating inflammasome assembly remains ill defined. Here, we demonstrate that NLRP3 inflammasome activators primarily converge on disruption of endoplasmic reticulum–endosome membrane contact sites (EECS). This defect causes endosomal accumulation of phosphatidylinositol 4-phosphate (PI4P) and a consequent impairment of endosome-to-*trans*-Golgi network trafficking (ETT), necessary steps for endosomal recruitment of NLRP3 and subsequent inflammasome activation. Lowering endosomal PI4P levels prevents endosomal association of NLRP3 and inhibits inflammasome activation. Disruption of EECS or ETT is sufficient to enhance endosomal PI4P levels, to recruit NLRP3 to endosomes and to potentiate NLRP3 inflammasome activation. Mice with defects in ETT in the myeloid compartment are more susceptible to lipopolysaccharide-induced sepsis. Our study thus identifies a distinct cellular mechanism leading to endosomal NLRP3 recruitment and inflammasome activation.

Inflammasomes are cytosolic multimeric protein complexes that play critical roles in innate immune responses to pathogens and damage-associated signals. The NLRP3 inflammasome is unique, as it is capable of detecting a remarkable variety of danger signals and it is therefore broadly implicated in different inflammatory diseases[1–4]. NLRP3 inflammasome activation requires two principal steps, priming through Toll-like or cytokine receptor signaling resulting in robust expression of the inflammasome components and their assembly upon exposure to NLRP3 inflammasome activating factors. Upon activation, NLRP3 oligomerizes and recruits the apoptosis-associated speck-like

(ASC) adapter and pro-caspase-1, leading to self-activation of caspase-1. Active caspase-1 cleaves pro-interleukin-1β (IL-1β) and pro-IL-18 into their mature forms and also cleaves gasdermin D, triggering a form of pro-inflammatory cell death called pyroptosis.

Depending on the nature of the activating stimuli, several mechanisms can lead to assembly of the NLRP3 inflammasome. Most of these mechanisms trigger potassium efflux to activate the NLRP3 inflammasome[5,6]. Yet, a primary sensing mechanism of the NLRP3 inflammasome receptor remains largely elusive. A recent study showed that NLRP3 recruitment to vesicles containing PI4P was important for its activation[7].

[1]Institut de génétique et de biologie moléculaire et cellulaire, Illkirch, France. [2]Centre national de la recherche scientifique, UMR7104, Illkirch, France. [3]Institut national de la santé et de la recherche médicale, U964, Illkirch, France. [4]Université de Strasbourg, Illkirch, France. [5]Telethon Institute of Genetics and Medicine, Pozzuoli, Italy. [6]Department of Molecular Medicine and Medical Biotechnology, University of Napoli Federico II, Medical School, Naples, Italy. [7]Department of Experimental Diabetology, German Institute of Human Nutrition Potsdam-Rehbruecke, Nuthetal, Germany. [8]Neurosciences and Cellular and Structural Biology Division, Eunice Kennedy Shriver National Institute of Child Health and Human Development, National Institutes of Health, Bethesda, MD, USA. [9]Laboratoire de biochimie et de biologie moléculaire, Nouvel Hôpital Civil, Strasbourg, France. [10]These authors contributed equally: Zhirong Zhang, Rossella Venditti. ✉e-mail: zhirong.zhang@igbmc.fr; dematteis@tigem.it; romeo.ricci@igbmc.fr

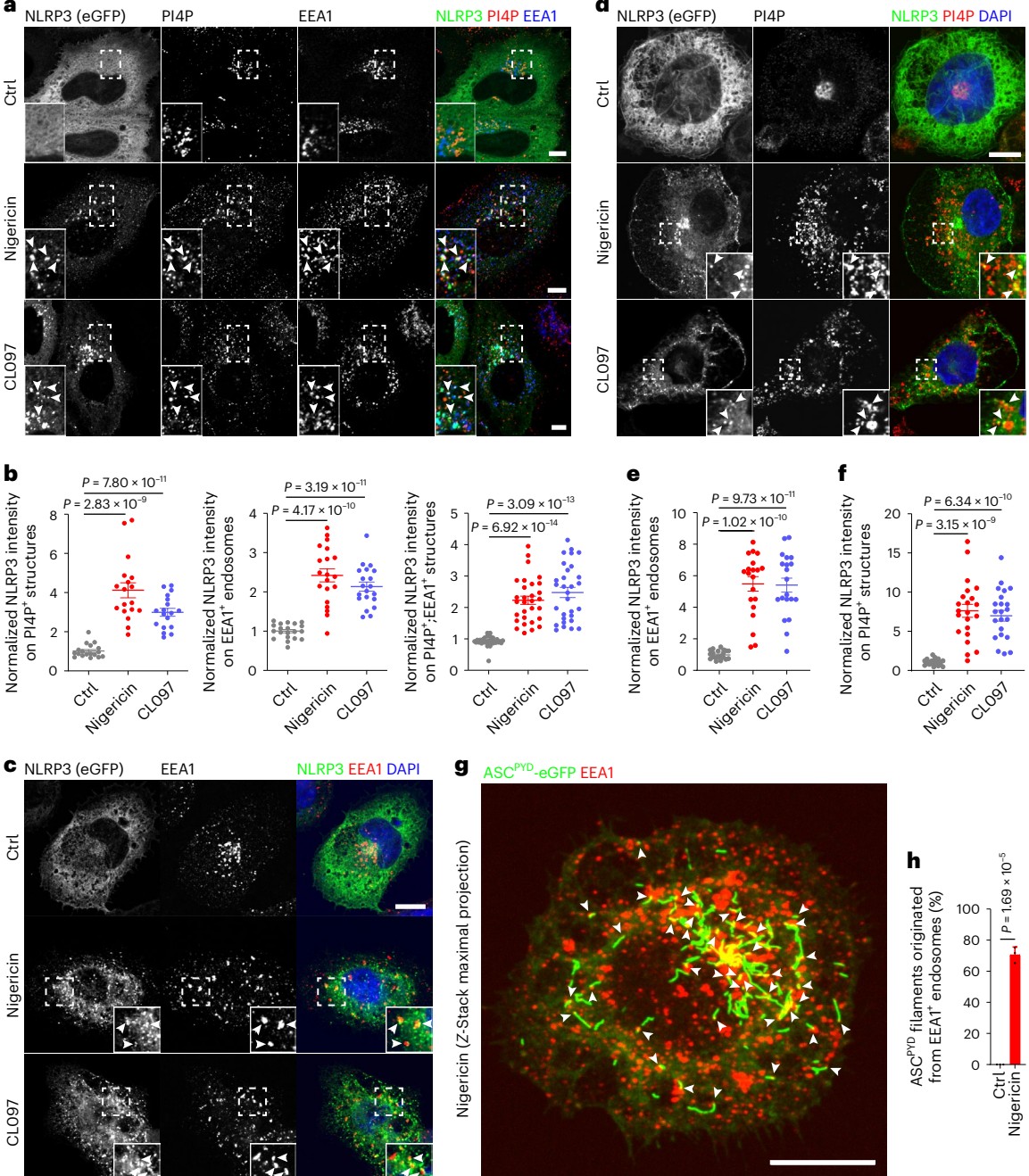

**Fig. 1 | NLRP3 inflammasome assembly originates from PI4P-enriched endosomes. a**, Confocal images of NLRP3-eGFP-expressing HeLa cells treated with vehicle (Ctrl), 15 μM nigericin or 45 μg ml⁻¹ CL097 for 60 min. Cells were co-stained with antibodies against PI4P and EEA1. Magnifications of areas in dashed squares are shown in the lower left corner. Arrowheads indicate EEA1-positive endosomes containing PI4P and NLRP3-eGFP. Scale bar, 10 μm. **b**, Quantification of relative intensity of NLRP3-eGFP in PI4P⁺-, EEA1⁺- and PI4P⁺;EEA1⁺ structures in experiments shown in **a**. Mean ± s.d., *N* = 3, *n* = 20 cells for each group. **c**, Confocal images of PMA-differentiated *ASC* KO THP-1 cells expressing NLRP3-eGFP treated with vehicle (Ctrl), 15 μM nigericin or 50 μM CL097 for 30 min. Cells were co-stained with an antibody against EEA1. DAPI was used to stain the nucleus. Magnifications of areas in dashed squares are shown in the lower right corner. Arrowheads indicate EEA1-positive endosomes containing NLRP3-eGFP. Scale bar, 10 μm. **d**, Confocal images of PMA-differentiated *ASC* KO THP-1 cells expressing NLRP3-eGFP treated with vehicle (Ctrl), 15 μM nigericin or 50 μM

CL097 for 30 min. Cells were co-stained with an antibody against PI4P. DAPI was used to stain the nucleus. Magnifications of areas in dashed squares are shown in the lower right corner. Arrowheads indicate PI4P-positive structures containing NLRP3-eGFP. Scale bar, 10 μm. **e**, Quantification of NLRP3-eGFP intensity on EEA1-positive endosomes in experiments shown in **c**. Mean ± s.d., *N* = 3, *n* = 20 cells for each group. **f**, Quantification of NLRP3-eGFP intensity on PI4P-positive vesicles in experiments shown in **d**. *N* = 3, *n* = 22 cells for each group. **g**, Z-stack maximal projection of confocal images of HeLa cells stably expressing ASC^PYD-eGFP and NLRP3 treated with 10 μM nigericin for 20 min. Cells were stained with an antibody against EEA1. DAPI was used to stain the nucleus. Arrowheads show ASC^PYD filaments originating from EEA1-positive endosomes. Scale bar, 0.5 μm. **h**, Quantification of ASC^PYD filaments originating from EEA1-positive endosomes in experiments shown in **g**. Mean of percentage ± s.d., *N* = 3. Data were analyzed with an unpaired two-sided *t*-test (**b**, **e**, **f**, **h**). Data shown in **a**, **c**, **d** and **g** are representative of three independent experiments.

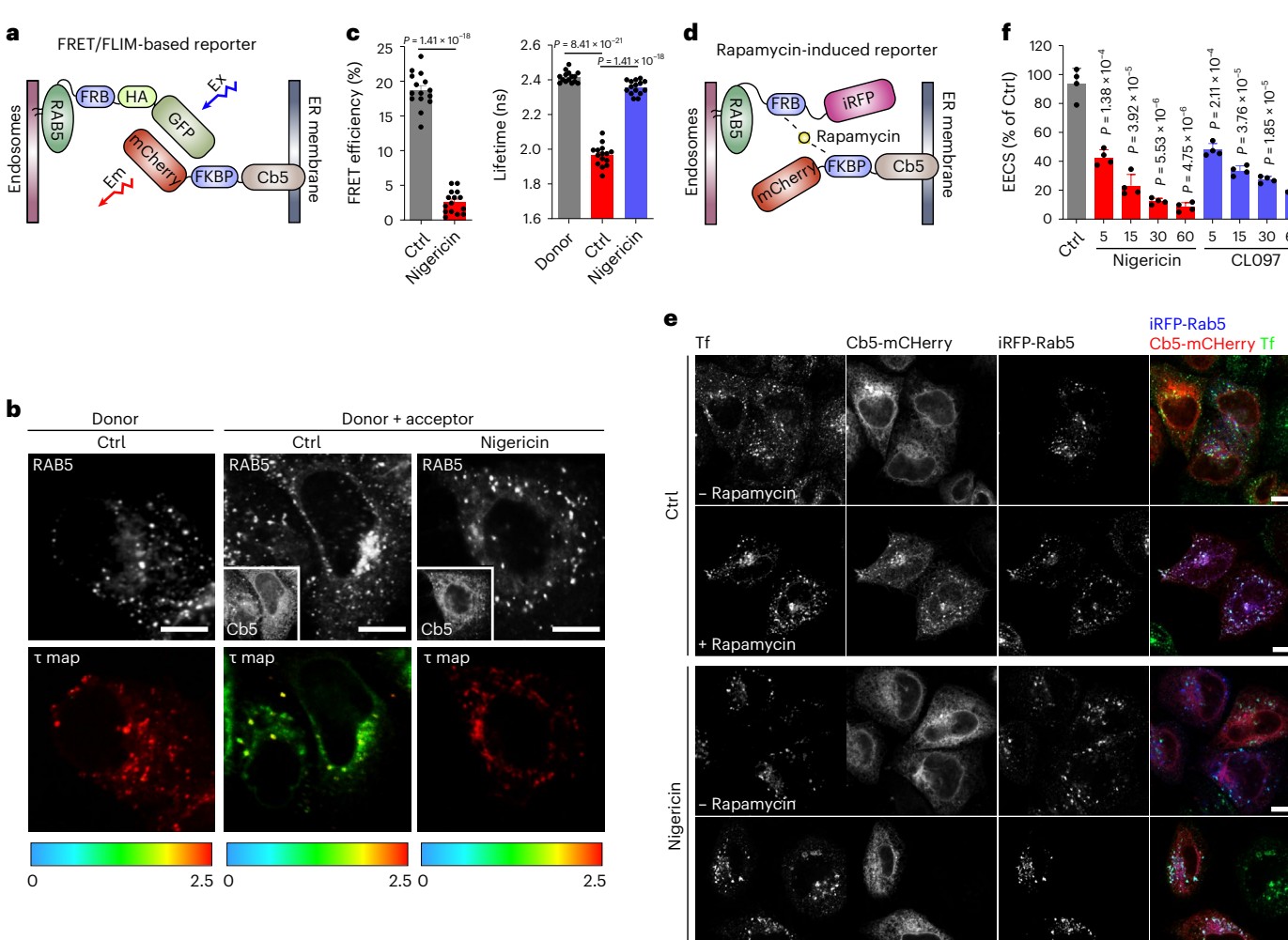

**Fig. 2 | NLRP3 activators disrupt EECS. a**, Schematic diagram of the FRET/FLIM-based reporter system for measurements of EECS. Ex, excitation of GFP; Em, emission of mCherry. **b**, Fluorescent (gray, upper) and FLIM (in color, lower) images of HeLa cells expressing RAB5-GFP (donor) alone or RAB5-GFP and mCherry-Cb5 (donor + acceptor) treated with vehicle (Ctrl) or 15 µM nigericin for 60 min. FLIM images show the spatial variation of the mean fluorescence lifetime ($\tau$ map) of Rab5-GFP (donor). $\tau$ values are represented by a pseudo-color scale ranging from 0 to 2.5 ns. Scale bar: 10 µm. **c**, Quantification of average donor lifetime (right) in experiments shown in **b**. FRET efficiency values (left) are calculated based on the RAB5-GFP donor lifetime. Mean ± s.d., $N = 3$, $n = 20$ cells for each group. **d**, Schematic diagram of the rapamycin-induced reporter

system for measurements of EECS. **e**, Confocal images of HeLa cells expressing RAB5-FRB-iRFP and mCherry-FKBP-Cb5 reporters treated with vehicle (Ctrl) or 15 µM nigericin for 60 min. Cells were incubated with (lower panels) or without (upper panels) 200 nM rapamycin for 4 min before fixation. Alexa-488-conjugated transferrin (Tf-Alexa-488, 5 µg ml⁻¹) was added to cells to stain endosomes. Scale bar, 10 µm. **f**, Quantification of EECS at indicated time points of nigericin and CL097 stimulation in experiments shown in **e**. Values are expressed as percentage of cells showing co-localization of Cb5 with RAB5. Mean ± s.d., $N = 4$, $n = 100$ cells for each group. Data were analyzed with an unpaired two-sided $t$-test (**c**, **f**). Data shown in **b** and **e** are representative of at least three independent experiments.

In this work, we find that the vesicles to which NLRP3 is recruited are of endosomal origin. We further demonstrate that endosomes accumulate PI4P, a required step for NLRP3 recruitment to endosomes and activation of the inflammasome at the endosomal membranes. PI4P accumulation in endosomes results from disruption of EECS and a subsequent defect in ETT that occurs in response to different NLRP3 inflammasome activators. Genetic disruption of EECS and ETT potentiates NLRP3 inflammasome activation. We propose that NLRP3 primarily senses distinct changes in the endosomal composition, including PI4P accumulation, in response to different NLRP3 activators.

## Results

### NLRP3 activators induce PI4P accumulation on endosomes

First, we assessed whether NLRP3 activators cause any change in intracellular PI4P levels or distribution using the PI4P probe GFP-P4C$_{SidC}$ and an antibody against PI4P in HeLa cells. We found that potassium

efflux-dependent nigericin as well as the potassium efflux-independent imiquimod-derived compound CL097 induced a fast and marked overall increase in PI4P levels in HeLa cells. Nigericin increased the levels of PI4P not only at the Golgi complex, the site where PI4P is particularly concentrated in non-treated cells, but also in sparse vesicles throughout the cytoplasm (Extended Data Fig. 1a,b). PI4P-positive cytoplasmic vesicles are early endosomes (EEs) as judged by transferrin loading and labeling with the EE marker SNX2 and EEA1 (Extended Data Fig. 1c–h). *ASC*-deficient THP-1 cells, a human acute leukemia monocytic cell line, and immortalized mouse bone marrow-derived macrophages (iBMDMs) lacking *ASC* also showed PI4P distribution to EEA1-positive vesicles upon nigericin or CL097 treatment (Extended Data Fig. 2a–d). The PI4P-positive vesicular compartment also contained PI3P (as judged by the PI3P probe, the FYVE domain of SARA), confirming its endosomal origin (Extended Data Fig. 1i,j). Thus, inflammasome activators generate an endosomal compartment with mixed phosphatidylinositol identity.

## Accumulation of PI4P on endosomes is required for recruitment and activation of NLRP3 in the endosomal compartment

Next, we asked whether NLRP3 is recruited to this compartment. Indeed, ectopically expressed enhanced green fluorescent protein-tagged NLRP3 (NLRP3-eGFP) acquired a punctate distribution co-localizing with PI4P-positive EEs upon nigericin or CL907 treatment, whereas it exhibited a diffuse distribution in non-treated cells (Fig. 1a,b). Furthermore, NLRP3-eGFP co-localized with EEA1-positive and PI4P-enriched vesicles in *ASC*-deficient THP-1 cells (Fig. 1c–f). Using the pyrin domain of ASC (ASC[PYD]) tagged with eGFP in HeLa cells as previously described[7], we confirmed that the tips of oligomerized ASC[PYD] filaments co-localized with the endosomal marker EEA1 (Fig. 1g,h), indicating that inflammasome assembly predominantly originates at endosomes.

Next, we addressed the source of increased endosomal PI4P in response to NLRP3 activators. Phosphatidylinositol 4-kinases, consisting of PI4KIIα, PI4KIIβ, PI4KIIIα and PI4KIIIβ, are important for the production of PI4P. First, we deleted *PI4KIIIβ*, which is the main source of PI4P at the Golgi, and assessed PI4P levels and endosomal recruitment of NLRP3 in HeLa cells in response to NLRP3 activators. Deletion of *PI4KIIIβ* depleted PI4P in the Golgi without altering endosomal PI4P levels and endosomal recruitment of NLRP3 in response to nigericin or CL097 (Extended Data Fig. 3a,b), excluding the possibility that increased endosomal PI4P is derived from the Golgi pool. Three kinases, PI4KIIα, PI4KIIβ and PI4KIIIβ, have been shown to locate to endosomes[8,9]. We therefore suspected functional redundancy of these kinases and combined depletion of *PI4KIIα* and *PI4KIIβ* with existing specific inhibitors. We found that the endosomal PI4P increase and the endosomal recruitment of NLRP3 induced by nigericin were dampened by the depletion of *PI4KIIα* and *PI4KIIβ* combined with the PI4KIIIβ inhibitor IN-9 in HeLa cells (Extended Data Fig. 3c,d). We then generated THP-1 cells lacking both *PI4KIIα* and *PI4KIIβ* using CRISPR/Cas9-mediated gene editing in combination with the PI4KIIIβ inhibitors IN-9 and IN-10 or the PI4KIIIα inhibitor GSK-A1. Deletion of *PI4KIIα* and *PI4KIIβ* in combination with IN-9 and IN-10 substantially reduced NLRP3 inflammasome activation in THP-1 cells. Western blotting revealed reduced release of mature IL-1β and cleaved caspase-1 to supernatants. Measurements of the cellular uptake of the live cell-impermeant nucleic acid dye Sytox Green revealed reduced pyroptosis. Finally, reduced IL-1β secretion was confirmed using an enzyme-linked immunosorbent assay (ELISA). However, deletion in combination with GSK-A1 had no such effect (Extended Data Fig. 3e,f). These data corroborated the importance of PI4P for the activation of the NLRP3 inflammasome and identified kinases responsible for PI4P accumulation in endosomes.

## NLRP3 activators lead to PI4P accumulation on endosomes via disruption of EECS

As previously reported, an increase of PI4P in endosomes can be caused by disruption of the EECS due to the inability of endosomal PI4P to be transferred by lipid transfer proteins such as oxysterol binding protein (OSBP) to the endoplasmic reticulum (ER) where it is degraded by PI4P phosphatase SAC1[10]. Depletion of VAP proteins (VAPA and VAPB), key components of ER–Golgi and EECS, as well as OSBP induces increased endosomal PI4P[10]. We therefore wondered whether NLRP3 activators affect contact sites. To this end, we developed a Förster resonance energy transfer (FRET)-based assay with a donor fluorophore molecule (GFP) conjugated to the endosomal protein RAB5 that, when excited, transfer energy to an acceptor fluorophore molecule (mCherry) conjugated to the ER membrane protein Cb5 in a situation in which EECS are maintained (Fig. 2a). A significant decrease of the RAB5-GFP mean lifetime value ($\tau$) was observed in cells expressing RAB5-GFP and mCherry-Cb5 (donor and acceptor) compared with cells expressing RAB5-GFP alone (donor). This decrease was abolished by nigericin treatment, suggesting that EECS were disrupted (Fig. 2b,c). To corroborate this phenomenon, we then used a system allowing for chemically induced dimerization of the RAB5 and Cb5 reporter proteins fused with FRB and FKBP domains by short treatment with a low concentration of rapamycin[11,12], which stabilizes pre-existing physiological EECS without inducing artificial tethering between endosomes and ER (Fig. 2d). Rapamycin induced co-localization of RAB5-FRB-iRFP and mCherry-FKBP-Cb5 in control cells. In contrast, co-localization was significantly decreased in nigericin-treated cells (Fig. 2e,f). Similar results were obtained upon CL097 stimulation (Fig. 2f). Thus, NLRP3 inflammasome activators converge on disruption of EECS.

Disruption of EECS is accompanied by an increase in actin polymerization on endosomal membranes manifested by actin comets that propel EEs. Strikingly, nigericin, as well as CL097, indeed induced the formation of actin comets on EEs in HeLa cells (Extended Data Fig. 4a,b and Supplementary Videos 1 and 2). Importantly, some of these EEs propelled by actin comets also contained NLRP3-eGFP, thus indicating that this altered EE compartment represents the site where NLRP3 is recruited (Extended Data Fig. 4c and Supplementary Video 3). Formation of actin comets was confirmed in nigericin-stimulated and CL097-stimulated primary BMDMs (Extended Data Fig. 4d).

In line with impaired delivery of the endosomal PI4P to the ER-located SAC1 as a consequence of EECS disruption, nigericin-induced increase of PI4P on endosomes and endosomal recruitment of NRLP3 were largely insensitive to ectopic expression of SAC1. However, these parameters were dramatically reduced upon ectopic expression of the cytosolic (and endosome-associated) SAC2 phosphatase (Extended Data Fig. 5a,b). Consistent with this observation, SAC2 was also much more efficient than SAC1 in decreasing NLRP3 inflammasome activity in wild-type (WT) THP-1 cells stimulated with nigericin, CL097 or imiquimod R837 (Extended Data Fig. 5c,d). These findings suggest that PI4P accumulation in endosomes as a consequence of EECS disruption is important for NLRP3 inflammasome activation.

---

**Fig. 3 | Disruption of EECS and impaired endosomal PI4P transport potentiate NLRP3 inflammasome activation. a**, Confocal imaging of WT, *VAP* dKO and *OSBP* KD HeLa cells expressing NLRP3-eGFP. Magnifications of areas in dashed squares are shown in the lower left corner. Arrowheads indicate EEA1-positive endosomes containing PI4P and NLRP3-eGFP. Scale bar, 10 μm. **b**, Quantification of percentage (mean ± SD) of NLRP3-expressing cells containing NLRP3 puncta (left) and number of NLRP3 puncta (right) in experiments shown in **a**. Left, $N = 4$, $n = 100$ cells; right, $N = 3$, $n = 30$ cells for each group. **c**, Immunoblotting of supernatants and cell lysates from WT and *VAPA/VAPB* double KO (*VAP* dKO) THP-1 cells treated or not treated with 1 μg ml$^{-1}$ LPS for 2 h in the presence of DMSO, 10 μM MCC950 or 20 mM KCl. Antibodies recognizing both p45 and p20 fragments of caspase-1 (CASP1), p31 and p17 fragments of IL-1β, VAPA and VAPB were used. **d**, Immunoblotting of supernatants and cell lysates from THP-1 cells expressing sgRNAs targeting *GFP* (sg*GFP*) or *OSBP* (sg*OSBP*) treated or not treated as indicated for **c**. Antibodies as described for **c** and against OSBP were used. **e**, Immunoblotting of supernatants and cell lysates from WT and *NLRP3* KO THP-1 cells expressing sgRNAs targeting *GFP* (sg*GFP*) or *OSBP* (sg*OSBP*). Cells were treated with vehicle, 1 μg ml$^{-1}$ LPS or 1 μg ml$^{-1}$ Pam3CSK4 for 2 h. Antibodies recognizing both p45 and p20 fragments of caspase-1, NLRP3 and OSBP were used. **f**, Immunoblotting of supernatants and cell lysates from WT THP-1 cells expressing GFP and *VAP* dKO THP-1 cells expressing GFP, WT hVAPA (VAPA WT) or K94D/M96D mutant hVAPA (VAPA mut). Cells were treated with 1 μg ml$^{-1}$ LPS for 2 h. Antibodies as described for **c** were used. Tubulin was chosen as a loading control for all immunoblots. **g, h, i, j**, Cellular uptake of Sytox Green and ELISA analysis of cytokine secretion in experiments are shown in **c**, **d**, **e** and **f**, respectively. Mean ± s.d., $N = 3$. \*\*\*$P < 0.001$; ns, not significant. Data were analyzed with an unpaired two-sided *t*-test (**b, g, h, i, j**). Data shown in **a, c, d, e** and **f** are representative of at least three independent experiments.

---

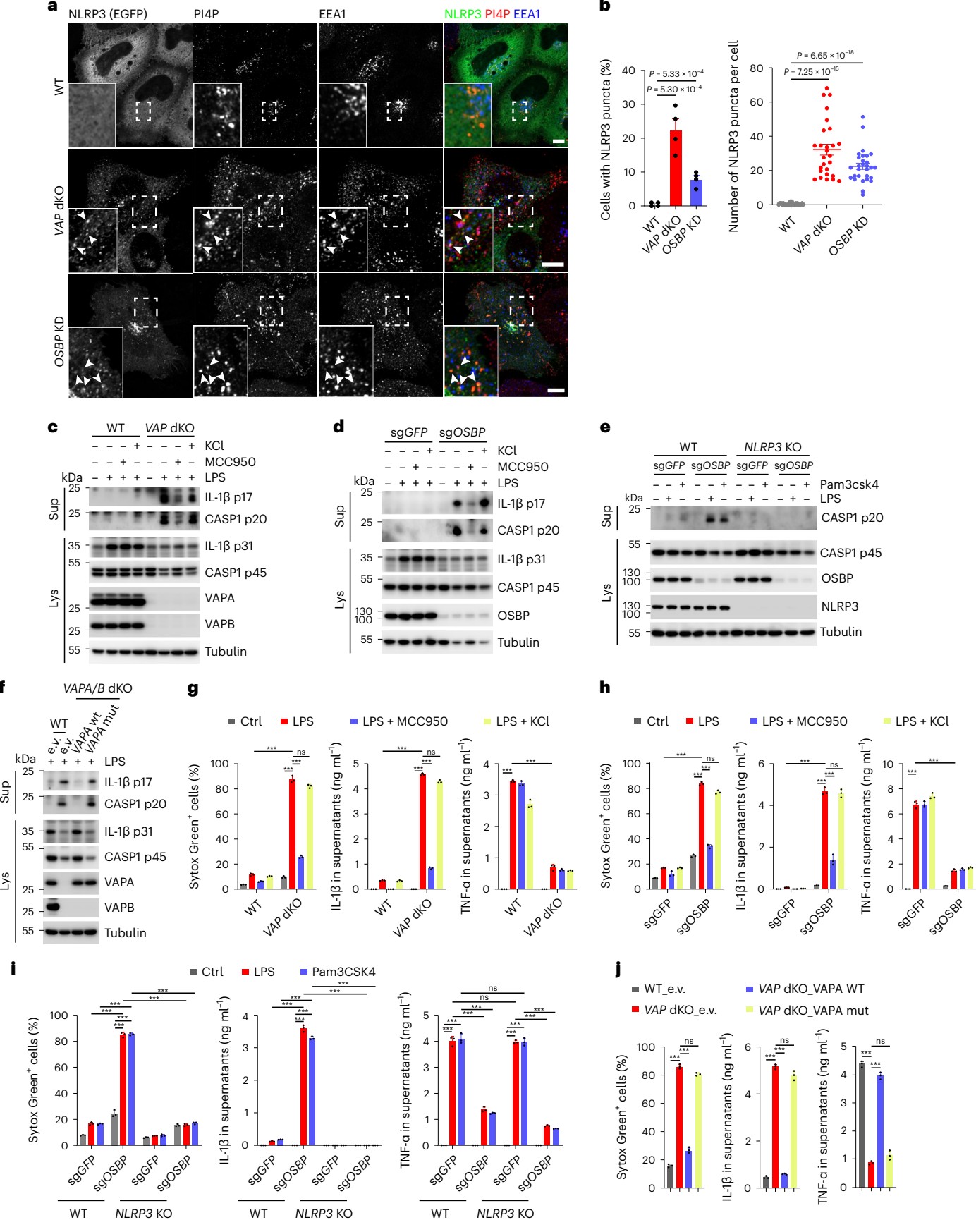

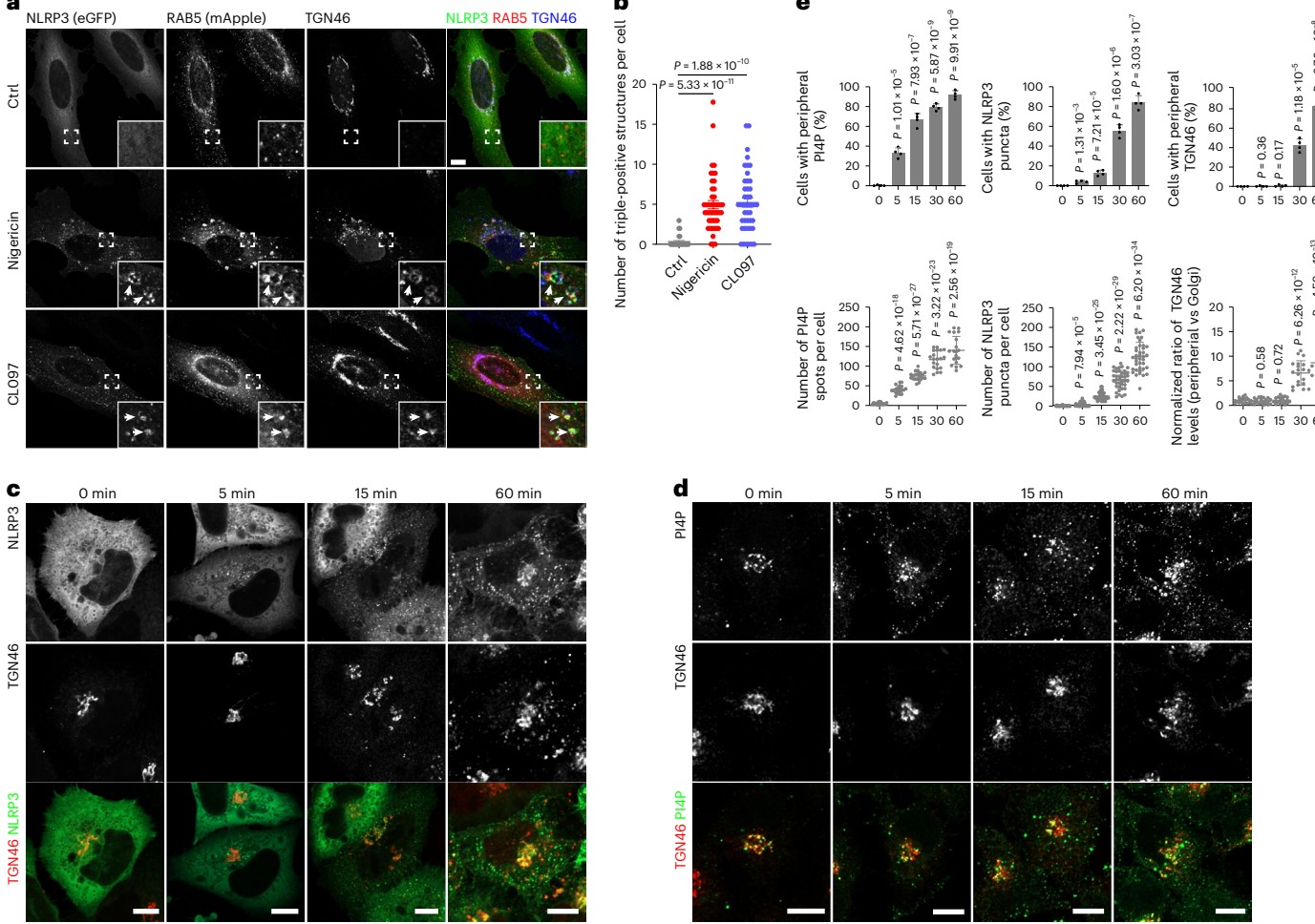

**Fig. 4 | NLRP3 activators result in retention of TGN46 on endosomes.**
**a**, Confocal images of HeLa cells stably expressing eGFP-tagged NLRP3 and mApple-tagged RAB5A. Cells were treated with vehicle (Ctrl), 10 μM nigericin for 40 min or 45 μg ml⁻¹ CL097 for 80 min and stained with an antibody against TGN46. DAPI was used to stain the nucleus. Magnifications of areas in dashed squares are shown in the lower right corner. Arrowheads indicate RAB5-positive endosomes containing TGN46 and NLRP3-eGFP. Scale bar, 10 μm. **b**, Quantification of the number of RAB5-positive endosomes containing NLRP3-eGFP and TGN46 in experiments shown in **a**. Mean ± s.d., N = 3, n = 34 cells for vehicle (Ctrl) group; n = 47 cells for nigericin-treated group and n = 44 for CL097-treated group. **c**, Confocal images of HeLa cells expressing NLRP3-eGFP before and after treatment with 15 μM nigericin for the indicated times. Cells were

stained with an antibody against TGN46. Scale bar, 10 μm. **d**, Confocal images of HeLa cells expressing HA-tagged TGN46 (TGN46-HA) before and after treatment with 15 μM nigericin for the indicated times. Cells were stained with antibodies against PI4P and HA tag. Scale bar, 10 μm. **e**, Upper panels: Quantification of temporally resolved nigericin-treated HeLa cells showing peripheral PI4P (left), NLRP3 puncta formation (middle) and TGN46 dispersion (right) in experiments shown in **c** and **d**. Mean of percentage ± s.d., N = 4, n = 100 cells for each group. Lower panels: Quantification of numbers of peripheral PI4P spots (left), NLRP3 puncta (middle) and TGN46 peripheral distribution (right) in each cell during nigericin treatment in experiments shown in **c** and **d**. Mean ± s.d., N = 3, n = 20 cells for each group. Data were analyzed with an unpaired two-sided t-test (**b**, **e**). Data shown in **a**, **c** and **d** are representative of at least three independent experiments.

## Disruption of EECS and impaired transfer of endosomal PI4P to ER potentiates NLRP3 inflammasome activation

Next, we asked whether disrupting the EECS or interfering with PI4P exchange at the level of EECS was per se sufficient to trigger NLRP3 activation. To this end, we assessed NLRP3 localization in *VAPA/B* double knockout (dKO)[10] and *OSBP*-depleted HeLa cells. We found that NLRP3-eGFP is constitutively localized to PI4P-enriched endosomes in both *VAPA/B* dKO *and OSBP*-depleted cells in the absence of inflammasome activators (Fig. 3a,b). Next, we generated *VAPA/B* dKO and *OSBP* KO THP-1 cells to assess NLRP3 inflammasome activity compared with WT cells using CRISPR/Cas9-mediated gene editing. *VAPA/B* dKO and *OSBP* KO THP-1 cells showed prominent pyroptotic cell death, secretion and cleavage of IL-1β and caspase-1 upon priming by lipopolysaccharide (LPS) (a TLR4 ligand) or Pam3CSK4 (a TLR1/TLR2 ligand), whereas priming alone had no such effect in corresponding WT control cells (Fig. 3c–e,g–i). These effects were independent of potassium efflux,

as increase of extracellular potassium levels did not show any differences, indicating that disruption of EECS in response to nigericin occurs downstream of potassium efflux (Fig. 3c,d,g,h). However, they were dependent on NLRP3, as inhibition or deletion of NLRP3 abolished these effects (Fig. 3c–e,g–i). Interestingly, tumor-necrosis factor-α (TNF-α) release was dampened from *VAPA/B* dKO and *OSBP* null THP-1 cells compared with levels from control cells (Fig. 3g–i). Re-expression of VAPA in *VAPA/B* dKO THP-1 cells abolished enhanced susceptibility to LPS priming (Fig. 3f,j). In contrast, the VAPA K94D/M96D mutant devoid of FFAT motif binding that is required for the formation of contact sites did not abolish increased susceptibility to LPS priming (Fig. 3f,j).

Interestingly, using a rapamycin-induced reporter dimerization assay[12], nigericin and CL097 also disrupted ER–*trans*-Golgi network (TGN) contact sites (Extended Data Fig. 6a,b). ER–TGN contact sites have also been shown to be VAPA/B-dependent[12]. However, depletion of *ORP10*, which is specific for the maintenance of ER–TGN contact

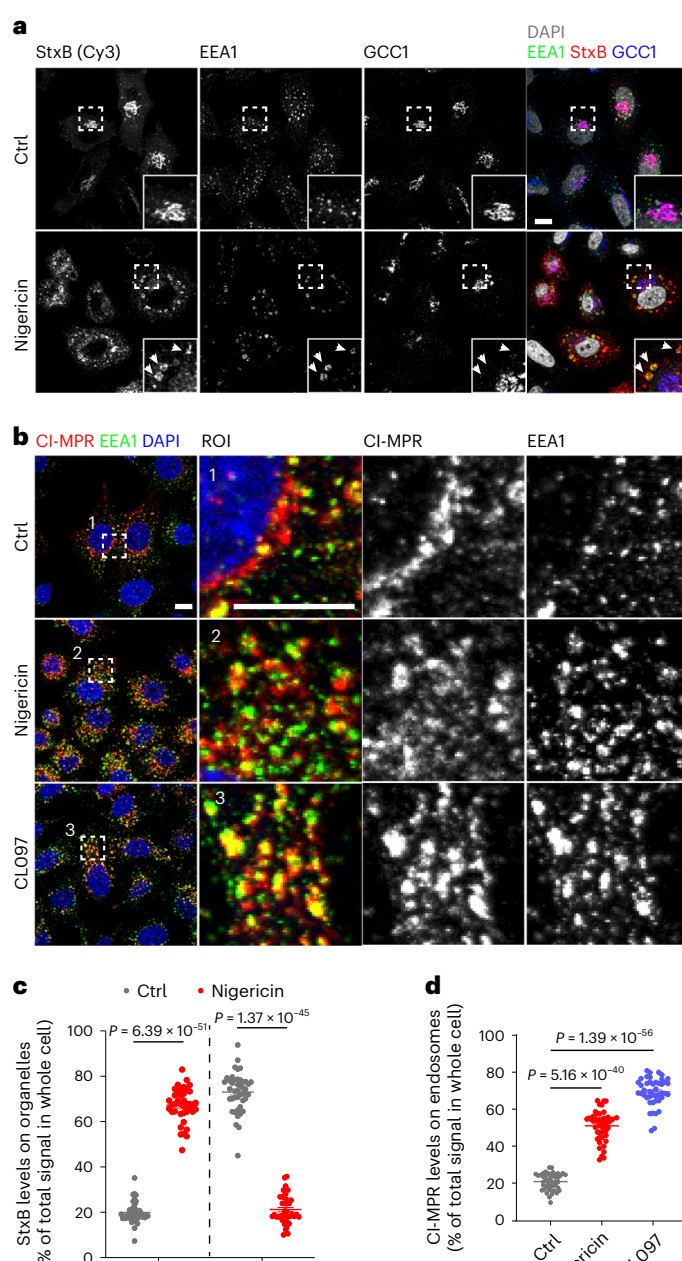

**Fig. 5 | NLRP3 activators disrupt ETT. a,** Internalization of StxB in HeLa cells. Cells were treated or not treated with 10 μM nigericin. Cy3-conjugated StxB was added into culture medium 5 min after the addition of nigericin. Cells were fixed at 40 min after incubation with Cy3-conjugated StxB. Cells were stained with antibodies against GCC1 and EEA1. DAPI was used to stain the nucleus. Magnifications of areas in dashed squares are shown in the lower right corner. Arrowheads indicate StxB retained on EEA1-positive endosomes. Scale bar, 10 μm. **b,** Internalization of CI-MPR antibody in HeLa cells. Cells were treated or not treated with 10 μM nigericin or 45 μg ml⁻¹ CL097. An antibody recognizing CI-MPR was added into the culture medium 5 min after nigericin or 20 min after CL097 treatment. Cells were fixed 40 min after the incubation with the CI-MPR antibody and were stained with an antibody against EEA1. DAPI was used to stain the nucleus. Magnifications of areas in numbered dashed squares are shown in separate numbered images. Scale bar, 10 μm. **c,** Quantification of StxB levels on endosomes and Golgi in experiments as described for **a.** Mean ± s.d., *N* = 3, *n* = 43 cells for control (ctrl) group and *n* = 40 cells for nigericin-treated group. **d,** Quantification of CI-MPR levels on endosomes in experiments in **b.** Mean ± s.d., *N* = 3, *n* = 47 cells for control group; *n* = 48 cells for nigericin-treated group and *n* = 44 for CL097-treated group. Data were analyzed with an unpaired two-tailed *t*-test (**c, d**). Data shown in **a** and **b** are representative of at least three independent experiments.

sites[12], did not show the same effect as depletion of *VAPA/B* or *OSBP* in response to LPS priming (Extended Data Fig. 6c,d), excluding a major role of disruption of ER–TGN contact sites. Overall, these data suggest that disruption of EECS is sufficient to potentiate NLRP3 inflammasome activation.

## NLRP3 activators impair ETT, resulting in endosomal retention of TGN-destined cargos

The consequences of the endosomal PI4P increase and of increased actin polymerization include ineffective endosome fission and impairment of ETT with endosomal retention of TGN-destined cargos[10] such as TGN46, which cycles between plasma membrane, endosomes and TGN[13–15]. Indeed, nigericin impaired the trafficking of TGN46 from the EEs to the TGN, thus causing re-localization of TGN46 from the Golgi area to scattered RAB5-positive EEs, and at the same time induced the association of NLRP3-eGFP to these vesicles in HeLa cells (Fig. 4a,b). The re-localization of TGN38 (mouse orthologue of human TGN46) into scattered endosomes was confirmed in primary BMDMs in response to nigericin or CL097 (Extended Data Fig. 7a,b).

Next, we dissected the temporal sequence of events induced by nigericin, that is, peripheral PI4P increase, endosomal recruitment of NLRP3 and peripheral redistribution of TGN46. We found that endosomal PI4P accumulation and NLRP3 recruitment preceded peripheral redistribution of TGN46, indicating that the latter is a consequence of increased endosomal PI4P that hampers its ETT (Fig. 4c–e). In line with this conclusion, the distribution of GRIP domain-containing TGN proteins, which do not undergo ETT[16], such as the golgins GCC1/GCC88, GCC2/GCC185, Golgin97 and Golgin245/p230, was not affected by treatment with nigericin or CL097, excluding defects in overall TGN integrity (Extended Data Figs. 7c–f and 8a–f).

To further address the extent to which NLRP3 inflammasome activators affect ETT, we examined their effect on the retrograde trafficking of Cy3-conjugated Shiga toxin subunit B (StxB)[17,18]. We observed that in control cells, the majority of StxB localized to a GCC1-positive TGN compartment after 40 min of endocytosis, in line with efficient endosomal retrograde transport to the TGN (Fig. 5a,c and Extended Data Fig. 9a,b). In contrast, in nigericin-treated (Fig. 5a,c) or CL097-treated cells (Extended Data Fig. 9a,b), StxB was mainly found in enlarged EEA1-positive EEs. As expected, StxB-enriched EEA1-positive vesicles were not positive for GCC1, GCC2 or p230, confirming their endosomal origin (Fig. 5a,c and Extended Data Fig. 9a–d). Importantly, StxB co-localized with TGN46 upon nigericin treatment, in line with the fact that TGN46 also transits along the ETT pathway (Extended Data Fig. 9e,f). These results indicated that treatment with nigericin or CL097 blocked ETT, leading to retention of StxB and TGN46 on endosomes. To corroborate this observation, we assessed the retrograde transport of another cargo: the cation-independent mannose-6-phosphate receptor (CI-MPR) as previously described[19,20]. We examined the internalization of an antibody recognizing the extracellular domain of CI-MPR[10] in control versus nigericin-treated or CL097-treated HeLa cells. In control cells, the CI-MPR antibody mainly localized to the TGN with only small amounts still at endosomes (Fig. 5b,d and Extended Data Fig. 9g,h). In contrast, most of the CI-MPR antibody was detected in endosomes but not at the TGN in nigericin-treated or CL097-treated cells (Fig. 5b,d and Extended Data Fig. 9g,h). These observations indicate that NLRP3 inflammasome activators impair ETT, resulting in endosomal retention of TGN cargos such as TGN46 and CI-MPR. These results align with a previous report that NLRP3-positive vesicles are positive for EEA1 and TGN46[7]. In that report, however, the authors concluded that EEA1 was missorted to Golgi-derived TGN46-positive vesicles. In contrast, our data support a model in which NLRP3-positive vesicles are of endosomal origin and that TGN46 is enriched on endosomes because of a defect in ETT.

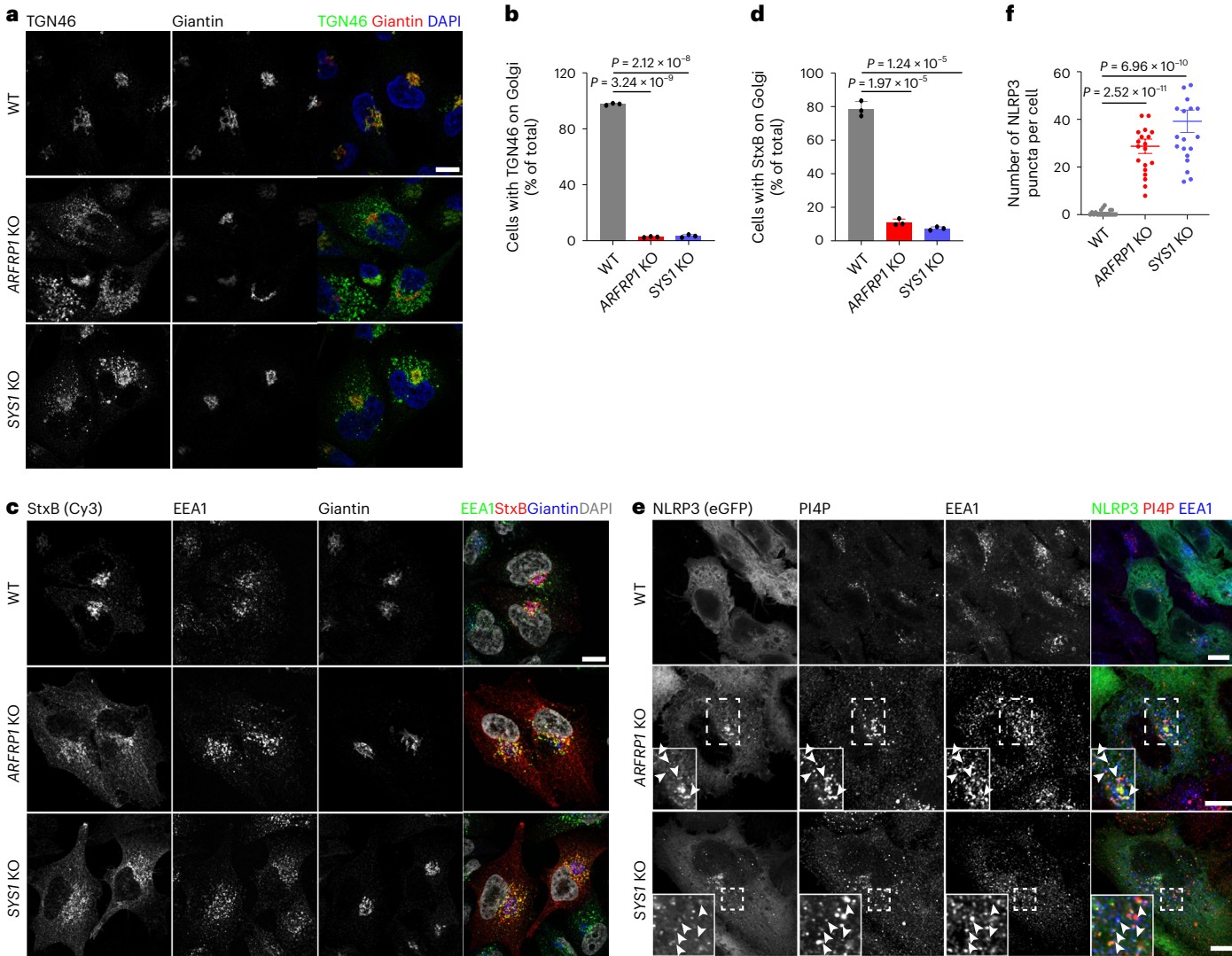

**Fig. 6 | Defective ETT contributes to PI4P accumulation on endosomes.**
**a**, Confocal images of WT, *ARFRP1* KO and *SYS1* KO HeLa cells. Cells were co-stained with antibodies against TGN46 and giantin. DAPI was used to stain the nucleus. Scale bar, 10 μm. **b**, Quantification of percentage of cells with TGN46 on the Golgi in experiments shown in **a**. Mean of percentage ± s.d., $N = 3$, $n = 100$ cells for each group. **c**, Internalization of StxB in WT, *ARFRP1* KO and *SYS1* KO HeLa cells. Cells were fixed at 40 min after incubation with Cy3-conjugated StxB and stained with antibodies against EEA1 and giantin. DAPI was used to stain the nucleus. Scale bar, 10 μm. **d**, Quantification of cells with StxB on the Golgi in experiments shown in **c**.

Mean of percentage ± s.d., $N = 3$, $n = 100$ cells for each group. **e**, Confocal images of WT, *ARFRP1* KO and *SYS1* KO HeLa cells expressing NLRP3-eGFP. Magnifications of areas in dashed squares are shown in the lower right corner. Arrowheads indicate EEA1-positive endosomes containing PI4P and NLRP3-eGFP. Scale bar, 10 μm. **f**, Quantification of NLRP3-eGFP puncta in experiments in **a**. Mean ± s.d., $N = 3$, $n = 20$ cells for each group. Data were analyzed with an unpaired two-sided *t*-test (**b**, **d**, **f**). Data shown in **a**, **c** and **d** are representative of at least three independent experiments.

## Disruption of ETT potentiates NLRP3 inflammasome activation

Next, we wondered whether defective ETT is a driving event in NLRP3 inflammasome activation. We targeted ETT through genetic means and assessed the effects of this manipulation on NLRP3 inflammasome activation. Both ADP ribosylation factor related protein 1 (ARFRP1) and SYS1 are critical regulators of TGN tethering factors, including long coiled-coil tethers of the golgin family and the heterotetrameric complex GARP, which mediates tethering and fusion of endosome-derived transport carriers with the TGN membranes[21]. In line with a previous study[21], deletion of *ARFRP1* or *SYS1* disrupted retrograde ETT of StxB toxin and TGN46 (Fig. 6a–d). In these cells, we found that NLRP3-eGFP was constitutively localized to endosomes in the absence of inflammasome activators, similar to those observed in *VAPA/B* dKO and *OSBP*-depleted cells shown above (Fig. 6e,f).

Similarly, enhanced endosomal PI4P levels were observed in these cells (Fig. 6e,f). We further deleted *ARFRP1* and *SYS1* in THP-1 cells using CRISPR/Cas9-mediated gene editing. Gene disruption was validated by immunoblotting or Sanger sequencing (Supplementary Fig. 1a–c). In line with a previous report[21], deletion of *SYS1* reduced protein levels of ARFRP1 (Supplementary Fig. 1c). Deletion of *ARFRP1* or *SYS1* did not affect expression of the inflammasome components, NLRP3, ASC, caspase-1 and NEK7 (Supplementary Fig. 1c). However, priming through stimulation with LPS or Pam3CSK4 was sufficient to induce lytic cell death of *ARFRP1* KO and *SYS1* KO THP-1 cells (Fig. 7a), accompanied by the cleavage and secretion of caspase-1 and IL-1β (Fig. 7b,c). In contrast, LPS or Pam3CSK4 alone did not evoke such a response in parental WT cells (Fig. 7a–c). LPS-induced lytic cell death, cleavage and secretion of caspase-1 and IL-1β in KO cells were dependent on NLRP3, as these effects were abolished in the presence of the NLRP3 inhibitor MCC950

(Fig. 7b,c) or by deletion of *NLRP3* (Fig. 7d). Interestingly, analogous to findings in *VAPA/B* dKO and *OSBP* KO cells, secretion of TNF-α was lower from *ARFRP1* KO and *SYS1* KO cells than its release from control cells in response to LPS priming (Fig. 7c). LPS treatment induced the assembly of mature inflammasome in these KO cells as evidenced by the formation of ASC specks (Fig. 7e,f). LPS-induced NLRP3 inflammasome activation in KO cells was independent of potassium efflux, as high extracellular potassium levels did not affect pyroptotic cell death, cleavage and secretion of caspase-1 and IL-1β induced by LPS (Fig. 7b,c). Furthermore, re-expression of WT ARFRP1 or constitutively active ARFRP1 Q79L, but not inactive ARFRP1 T31N or Flag-tagged SYS1 abrogated LPS-induced NLRP3 inflammasome activation in *ARFRP1* KO THP-1 cells (Fig. 7g,h), whereas only re-expression of Flag-tagged SYS1, but not ARFRP1 WT, Q79L or T31N, prevented LPS-induced NLRP3 inflammasome activation in *SYS1* KO THP-1 cells (Fig. 7g,h). These results are in line with previous studies showing that SYS1 and ARFRP1 are both critical components of TGN tethering[21,22]. Activation of other inflammasomes, including the NLRC4 or AIM2 inflammasome, was not affected in *ARFRP1* KO and *SYS1* KO cells (Fig. 7i). Of note, cells depleted of RAB7 (RAB7A and RAB7B), which is critical for late endosome maturation and lysosome biogenesis, did not show the same effect as observed in *ARFRP1* or *SYS1* KO cells (Supplementary Fig. 2a,b), excluding the major contribution of the endosome–lysosome transport pathway to NLRP3 inflammasome activation. From these experiments, we concluded that perturbation of the TGN tethering machinery was sufficient to activate the NLRP3 inflammasome in response to priming in THP-1 cells in a potassium efflux-independent manner.

To further test whether this mechanism is conserved among species, we generated iBMDMs lacking *Arfrp1* or *Sys1* and tested NLRP3 inflammasome activation (Fig. 7j). As observed in THP-1 KO cells, both *Arfrp1* and *Sys1* KO iBMDMs showed IL-1β release in response to LPS priming in the absence of NLRP3 activators, whereas WT cells did not (Fig. 7k). However, KO iBMDMs did not undergo lytic cell death in response to LPS priming as observed in THP-1 cells. Yet, KO iBMDMs revealed dramatically enhanced NLRP3 inflammasome activation when treated with nigericin or CL097 compared with WT cells (Fig. 7k).

To assess the in vivo relevance of our findings, we generated mice lacking *Arfrp1* in the myeloid lineage by crossing *Arfrp1*-floxed mice to *LysM-cre* mice. Efficient deletion of *Arfrp1* in BMDMs was confirmed by immunoblotting (Fig. 8a). *Arfrp1* KO BMDMs displayed significantly enhanced caspase-1 activation, IL-1β cleavage, and IL-1β secretion in response to ATP, nigericin, CL097 or R837 treatment compared with control cells (Fig. 8a,b). NLRP3 inflammasome activation and IL-1β release is crucial in low-dose LPS-induced endotoxemia[23]. Thus, we next performed intraperitoneal injections of LPS using a sublethal dose. Strikingly, myeloid-specific *Arfrp1* KO mice were more susceptible to an LPS challenge compared with control mice, as most of the myeloid-specific *Arfrp1* KO mice died, whereas most of the control mice survived at such a dose of LPS (Fig. 8c). Serum IL-1β levels were substantially enhanced in myeloid-specific *Arfrp1* KO mice compared

with levels in control mice (Fig. 8d). TNF-α levels were also slightly increased in myeloid-specific *Arfrp1* KO mice (Fig. 8d). Increased serum IL-1β and susceptibility to LPS were dependent on NLRP3, as MCC950 reversed these effects in myeloid-specific *Arfrp1* KO mice (Fig. 8c,d). Enhanced levels of TNF-α were also dependent on NLRP3 (Fig. 8d), indicating that increased TNF-α release was most likely a consequence of pro-inflammatory activity of enhanced NLRP3 inflammasome activation. These results suggest that defective ETT drives NLRP3 inflammasome activation in vivo in mice.

## Discussion

The NLRP3 inflammasome receptor is rather exceptional, as it is capable of sensing a broad range of cellular stressors. We demonstrate here that NLRP3 inflammasome activators converge into disruption of EECS, endosomal PI4P accumulation and recruitment of NLRP3 to endosomes. We were able to dissect the temporal sequence of events in response to NLRP3 activators: the destabilization of membrane contact sites, the increase of endosomal PI4P and the endosomal recruitment of NLRP3 precedes impairment of ETT that eventually results in retention of TGN cargo on endosomes (Fig. 8e). It will be of great interest in the future to understand how NLRP3 inflammasome activators disrupt EECS. We have also demonstrated that accumulation of PI4P on endosomes is critical for NLRP3 activation. However, whether full-length NLRP3 directly binds PI4P on endosomes is still to be determined, as forced binding of NLRP3 with PI4P was not sufficient to induce inflammasome assembly[7]. Of note, a very recent study demonstrated that the inactive NLRP3 dodecamer was capable of binding negatively charged lipids including PI4P in vitro[24]. Given the importance of defective ETT in inflammasome activation, cargo retention in endosomes might be another important step. How ETT contributes to endosomal PI4P accumulation is to be further investigated. NLRP3 might thus detect other molecules (lipids or proteins) that accumulate on endosomes in response to TGN cargo retention.

Although disruption of EECS or ETT is sufficient to activate NLRP3 on endosomes in HeLa cells, priming through LPS is needed in EECS- or ETT-deficient THP-1 cells to trigger inflammasome activation. One possible explanation of this apparent discrepancy is that overexpression of NLRP3 in HeLa cells overrides the requirement for LPS-induced priming. This hypothesis is in line with the observations that nigericin alone without LPS priming is sufficient to trigger the formation of ASC[PYD] filaments in HeLa cells, whereas priming is needed for endogenous NLRP3 activation in THP-1 cells. As NLRP3 is constitutively expressed in THP-1 cells, we assume that LPS-induced specific post-transcriptional modifications in NLRP3 are needed to allow NLRP3 translocation to endosomes. It is tempting to speculate that one of these modifications might favor conformational changes in NLRP3 necessary for PI4P binding. As for BMDMs, priming is needed to express NLRP3 in the first place.

Endosomes have emerged as signaling hubs involved in many cellular processes, including innate immune responses[25]. For instance, Toll-like receptors (TLRs) such as TLR3, TLR4, TLR7, TLR8, TLR9 and

**Fig. 7 | Defective ETT promotes NLRP3 inflammasome activation. a**, Cellular uptake of Sytox Green in THP-1 cells with indicated genotypes treated or not treated with 1 µg ml⁻¹ LPS or Pam3CSK4 for 2 h. Mean ± s.d., *N* = 3. ****P* < 0.001. **b**, Immunoblotting of supernatants and cell lysates from THP-1 cells with indicated genotypes treated or not treated with 1 µg ml⁻¹ LPS in the absence or presence of 10 µM MCC950 or 20 mM KCl for 2 h. **c**, Cellular uptake of Sytox Green and ELISA analysis of cytokine secretion in experiments shown in **b**. Mean ± s.d., *N* = 3. ****P* < 0.001; ns, not significant. **d**, Immunoblotting of culture supernatants and lysates from THP-1 cells. THP-1 cells with indicated genotypes expressing sgRNAs against *ARFRP1* or *SYS1* were treated with 1 µg ml⁻¹ LPS for 2 h. **e**, Confocal images of THP-1 cells with indicated genotypes treated with 1 µg ml⁻¹ LPS for 2 h stained with an antibody against ASC. Arrowheads indicate ASC specks. Scale bar, 20 µm. **f**, Quantification of ASC speck-containing cells in experiments shown in **g**.

Mean ± s.d., *N* = 3, *n* = 100 cells for each group. ****P* < 0.001. **g**, Immunoblotting of supernatants and lysates from THP-1 cells with indicated genotypes expressing GFP, Flag-SYS1, WT ARFRP1, T31N or Q79L mutant ARFRP1 treated with 1 µg ml⁻¹ LPS for 2 h. **h**, Cellular uptake of Sytox Green and ELISA analysis of cytokine secretion in experiments shown in **g**. Mean ± s.d., *N* = 3. ****P* < 0.001; ns, not significant. **i**, Immunoblotting of supernatants and lysates from THP-1 cells with indicated genotypes treated or not treated with FLn-Needle plus anthrax protective antigen or transfected with poly(dA:dT) for 4 h. **j**, Validation of generation of *Arfrp1* KO and *Sys1* KO iBMDMs. **k**, ELISA of IL-1β in supernatants from iBMDMs with indicated genotypes. After LPS for 4 h, cells were treated with 7.5 µM nigericin or 50 µM CL097 for 30 min. Mean ± s.d., *N* = 3. ****P* < 0.001. Data were analyzed with an unpaired two-sided *t*-test (**a**, **c**, **f**, **h**, **k**). Data shown in **b**, **d**, **e**, **g**, **i** and **j** are representative of at least three independent experiments.

TLR13 are located on endosomal membranes[25]. Endosomal location of TLR4 is critical for TRIF-mediated signaling[26]. The endosomal location of other TLRs is critical for their functions in detecting cognate microbial ligands[27–31]. In this study, we discovered that distinct changes in the composition of endosomes is a mark for activation of innate immunity mediated by NLRP3. Our findings thus define an unrecognized role of the endosomal compartment in pattern recognition receptor-induced innate immune responses.

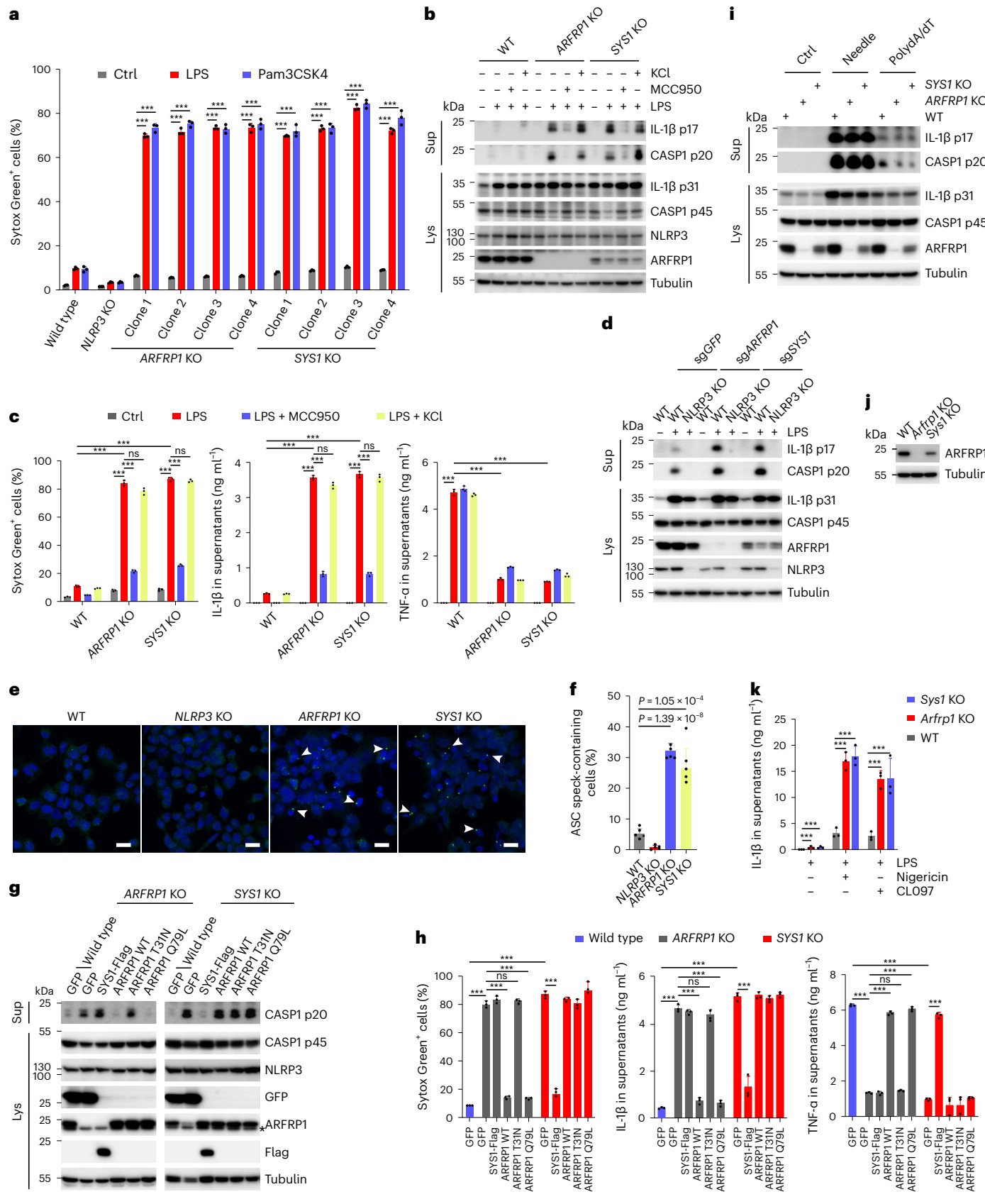

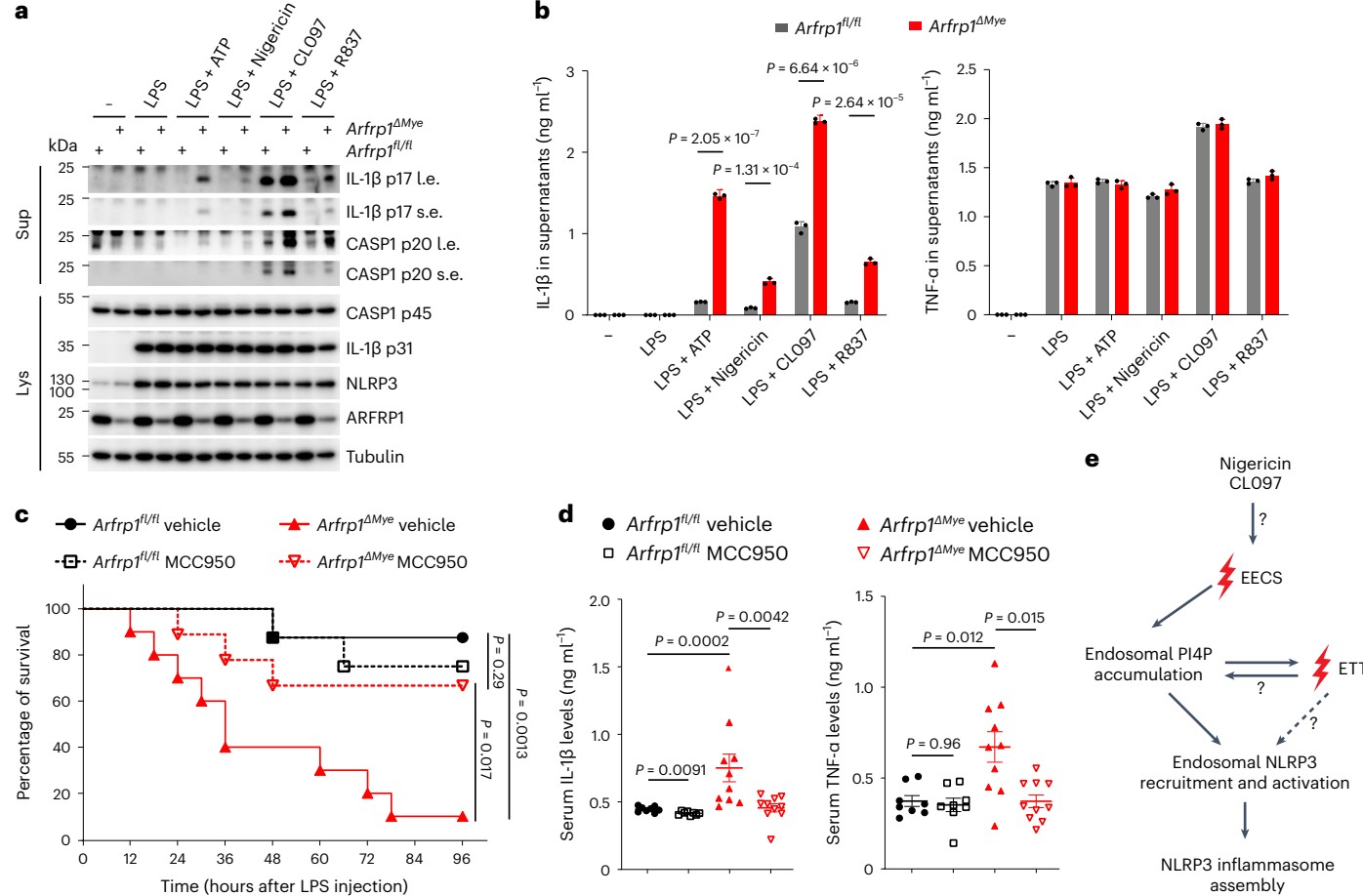

**Fig. 8 | Defective ETT promotes NLRP3 inflammasome activation in mice. a**, Immunoblotting of supernatants and lysates from primary BMDMs isolated from *Arfrp1^fl/fl* or *Arfrp1^fl/fl;LysM-Cre (Arfrp1^ΔMye)* mice. Isolated cells were primed with 1 μg ml⁻¹ LPS for 4 h, followed by treatment with 2 mM ATP, 5 μM nigericin, 50 μM CL097 or 50 μM R837 for 40 min as indicated. Antibodies recognizing both p45 and p20 fragments of caspase-1 (CASP1), p31 and p17 fragments of IL-1β, NLRP3 and ARFRP1 were used. An antibody against tubulin was used as a loading control. Short exposures (s.e.) and long exposures (l.e.) of IL-1β and CASP1 fragments are shown. **b**, ELISA assay of IL-1β and TNF-α in supernatants of isolated BMDMs from mice treated as described for **a**. Mean ± s.d., *N* = 3. Data were analyzed with an unpaired two-sided *t*-test. **c**, Survival curves of *Arfrp1^fl/fl* or *Arfrp1^fl/fl;LysM-Cre (Arfrp1^ΔMye)* mice injected with LPS (15 mg kg⁻¹) in the presence of vehicle or 50 mg kg⁻¹ MCC950. The survival of mice after LPS injection was monitored

every 6 h. *n* = 8 for vehicle-treated *Arfrp1^fl/fl* mice group; *n* = 8 for MCC950-treated *Arfrp1^fl/fl* mice group; *n* = 10 for vehicle-treated *Arfrp1^ΔMye* mice group and *n* = 9 for MCC950-treated *Arfrp1^ΔMye* mice group. Data were analyzed with a log-rank (Mantel−Cox) test. **d**, ELISA assay of IL-1β (left panel) and TNF-α (right panel) levels in serum collected from *Arfrp1^fl/fl* or *Arfrp1^fl/fl;LysM-Cre (Arfrp1^ΔMye)* mice at 3 h after peritoneal injections of LPS (15 mg kg⁻¹) in the presence of vehicle or 50 mg kg⁻¹ MCC950. Mean ± s.d., *n* = 8 for vehicle-treated *Arfrp1^fl/fl* mice group; *n* = 8 for MCC950-treated *Arfrp1^fl/fl* mice group; *n* = 10 for vehicle-treated *Arfrp1^ΔMye* mice group and *n* = 9 for MCC950-treated *Arfrp1^ΔMye* mice group. Data were analyzed with a two-sided Mann−Whitney *U*-test. **e**, Proposed model of NLRP3 inflammasome activation as described in Discussion. Data shown in **a** are representative of three independent experiments.

In conclusion, our study provides evidence for NLRP3 to primarily sense endosomal stress occurring through defective EECS and ETT.

## Online content

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

## Methods

### Mice
Mice with targeted alleles for *Arfrp1* (*Arfrp1*[fl/fl]) were described previously[32] and were provided by Annette Schürmann (German Institute of Human Nutrition Potsdam-Rehbruecke). *Arfrp1*[fl/fl] mice on C57BL/6J background were crossed with *LysM-Cre* mice to obtain myeloid-specific *Arfrp1* KO mice. Mice were housed under specific pathogen-free conditions with controlled temperature (19–23 °C) and humidity (40–60%) on a 12 h/12 h light/dark cycle with unrestricted access to water and standard laboratory chow. Maintenance and animal experimentation were in accordance with the local ethical committee (Com'Eth) in compliance with the European legislation on care and use of laboratory animals (La cellule AFiS (Animaux utilisés à des Fins Scientifiques): APAFIS#30865-2021040115389039 v3). No exclusion of animals used for experiments was performed. Healthy mouse littermates were chosen randomly according to their genotypes.

### Chemicals
Nigericin sodium salt (N7143), adenosine 5′-triphosphate disodium salt (ATP; A2383), StxB (SML0562), GSK-A1 (SML2453), rapamycin, phorbol 12-myristate 13-acetate (PMA; P8139), anthrax protective antigen (176905) and LPS from *Escherichia coli O55:B5* (L2880) were purchased from Sigma-Aldrich. Pam3CSK4 (tlrl-pms), imiquimod (tlrl-imq), CL097 (tlrl-c97), FLn-Needle (tlrl-ndl) and poly(dA:dT) (tlrl-patn) were obtained from InvivoGen. MCC950 (HY-12815), PI4KIIIβ-IN-9 (HY-19798) and PI4KIIIβ-IN-10 (HY-100198) were purchased from MedChemExpress. Sytox Green nucleic acid stain (S7020), Alexa-488-conjugated transferrin (T13342), Alexa-633-conjugated transferrin (T23362) and Alexa-568-conjugated phalloidin (A12380) were purchased from Thermo Fisher Scientific. Cy3-StxB was generated by conjugating Cy3 dye onto StxB according to the manufacturer's manual of Cy3 Ab Kit (Sigma-Aldrich, PA330000).

### DNA plasmids
pX330-U6-Chimeric_BB-CBh-hSpCas9 (42230), lentiCRISPR v2 (52961) and iRFP-FKB-Rab5 (51612) were purchased from Addgene. eGFP-tagged mouse *NLRP3* and eGFP-tagged human *SNX2* were cloned into pBOB empty vector using ligation-independent cloning (LIC) as described previously[33]. The pEGFP-C1-HA-FKB-RAB5a was generated using LIC/PIPE cloning, in which the fragment HA-FKB-RAB5 was amplified from iRFP-FKP-RAB5 using the following three primers to insert the HA tag by PCR:

HA x GFP fw1, AGATCTCGAGCTCAAGCTTCGAATTCAACCACCATG GTTTATCCATACGACGTTCCAGACTATGC;

HA_FRAP_fw2, CACCACCATGGTTTATCCATACGACGTTCCAGAC TATGCCGGAGGAGGTATGTGGCATGAAGGCCTGG;

RAB5 x GFP-rev, TGATCAGTTATCTAGATCCGGTGGATCCTTAC TAGTTACTACAACACTGACTCCTGGTTGGC.

The pEGFP-C1 plasmid was amplified with the following primers:
GFP-C (+), TAAGGATCCACCGGATCTAGATAACTGATCA;
GFP-C (−), GAATTCGAAGCTTGAGCTCGAGATC.

The ER–TGN contact sites reporter construct TGN46-FRB-HA-GFP-T2A-mCherry-FKBP-Cb5-pIRES-neo2 was generated as described in ref. [12]. The EECS reporter Cb5 (17 residues of the transmembrane domain of rat cytochrome b5)[34] was cloned in frame downstream of the mCherry-FKBP construct as described in ref. [12]. The PI4P probe GFP-SidC was a kind gift from Y. Mao (Cornell University). The PI3P probe GFP-FYVE-SARA was a kind gift from S. Corvera (University of Massachusetts Medical School). LifeAct-RFP was derived a construct from ref. [35]. pEGFP-C1 hVAPA was from Fabien Alpy (Institut de génétique et de biologie moléculaire et cellulaire, IGBMC). pEGFP-N1 hSYS1 and pEGFP-N1 hARFRP1 (WT, Q79L, T31N) were described previously[21]. Flag-tagged hSYS1, eGFP-tagged hARFRP1 (WT, Q79L, T31N), hVAPA (WT, K94D/M96D mutant), ASC[PYD] fused with carboxy-terminal eGFP, and hSAC1 and hSAC2 fused with amino-terminal mCherry were

amplified by PCR and were cloned into pBOB empty vector at the BamHI site. The pX330-P2A-eGFP/RFP plasmids were generated by inserting the P2A-eGFP/RFP sequence into EcoRI-digested pX330-U6-Chimeric_BB-CBh-hSpCas9 before the stop codon using LIC. Small guide RNA (sgRNA) sequences targeting human ARFRP1 (h*ARFRP1)*, h*SYS1*, h*VAPA*, h*VAPB*, h*ORP10*, h*RAB7A* and h*RAB7B* gene were inserted into Bbs-I-digested pX330-P2A-eGFP/RFP plasmids through ligation by T4 DNA ligase. sgRNA sequences targeting mouse *Arfrp1* (m*Arfrp1*), m*Sys1*, hOSBP, h*PI4KIIIβ*, h*PI4KIIα* and h*PI4KIIβ* were inserted into Bbs-I-digested lentiCRISPR v2 through ligation by T4 DNA ligase.

The sequences of sgRNAs used in this study are listed in Supplementary Table 1.

### Cell culture
All mammalian cells were cultured at 37 °C, 5% $CO_2$. Cell lines used in this study are not listed in the databases of commonly misidentified cell lines maintained by the International Cell Line Authentication Committee (ICLAC). THP-1 cells (American Type Culture Collection, ATCC) were grown in RPMI 1640 containing 10% fetal bovine serum, 10 mM HEPES, 2.5 g l⁻¹ glucose, 1 mM sodium pyruvate and gentamycin. HeLa cells were grown in DMEM (4.5 g l⁻¹ glucose) containing 10% fetal bovine serum, penicillin and streptomycin. HEK293T cells (ATCC, CRL-3216) obtained from the German Cancer Research Center (DKFZ Heidelberg) were grown in DMEM containing 1 g ml⁻¹ glucose, 10% fetal bovine serum, penicillin and streptomycin. *NLRP3* KO and *ASC* KO THP-1 cells were generated as previously described[33]. The RAB5-mApple stably expressing HeLa cell line was provided by Anne Spang (Biozentrum, University of Basel). The HeLa WT and HeLa *VAP* dKO cell lines[10] were a generous gift from Pietro De Camilli (Yale University School of Medicine). The iBMDM cell line was obtained from Eicke Latz (University of Bonn) and was grown in DMEM (4.5 g l⁻¹ glucose) containing 10% heat-inactivated fetal bovine serum, penicillin and streptomycin. Primary BMDMs were obtained by differentiating bone marrow progenitors from the tibia and femur in RPMI 1640 containing 50 ng ml⁻¹ recombinant hM-CSF (ImmunoTools, 11343113), 10% heat-inactivated fetal bovine serum, penicillin and streptomycin for 7 days. BMDMs and peritoneal macrophages were seeded 1 day before experiments using RPMI 1640 containing 10% heat-inactivated fetal bovine serum, penicillin and streptomycin. Cell cultures were negative for mycoplasma contamination. THP-1 cells and HEK293T cells have been authenticated using short random repeat (STR) performed by LGC Standards.

HeLa cells were transfected with plasmids using TransIT-LT1 (Mirus Bio) according to the manufacturer's instructions. Expression was maintained for 16–24 h before processing unless otherwise stated. For RNA interference, HeLa cells were treated with control short interfering RNA (siRNA) or OSBP, PI4KIIα and PI4KIIβ siRNA (50 nM) for 72 h using Oligofectamine (Thermo Fisher Scientific) by direct transfection. siRNA sequences used in this study are listed in Supplementary Table 1.

For treatment, BMDMs were primed with 1 μg ml⁻¹ LPS for 4 h, followed by the treatment of ATP, nigericin, imiquimod and CL097; THP-1 cells were differentiated by PMA (100 nM) treatment for 3 h followed by overnight incubation with fresh medium before experiments. To activate the NLRC4- and AIM2 inflammasome in *ARFRP1* KO and *SYS1* KO THP-1 cells, PMA-differentiated WT, *ARFRP1* KO and *SYS1* KO THP-1 cells were pre-treated with 10 μM MCC950 (to avoid spontaneous NLRP3 inflammasome activation in these cells) for 30 min, followed by treatment with LFn-Needle plus anthrax protective antigen (100 ng ml⁻¹ each) or transfected with poly(dA:dT) (double-stranded DNA, dsDNA) (1 μg ml⁻¹) for 4 h in the presence of 10 μM MCC950.

For the packaging of lentivirus, 12 μg of Lenti-mix (3 μg pVSVG, 3 μg pMDL and 3 μg pREV) plus 12 μg of the gene of interest expressing plasmid were transfected into HEK293T cells (10 cm plate) using Lipofectamine 2000. After 48 h, the supernatants were collected and filtered using 0.45 μm Millex-HV Syringe Filters and kept at −80 °C. BMDMs and THP-1 cells were infected in fresh medium containing

1 µg ml⁻¹ polybrene (Santa Cruz Biotechnology, sc-134220) and 25% lentivirus-contained supernatant.

## Gene disruption using the CRISPR/Cas9 genome editing system

To generate the *ARFRP1* KO, *SYS1* KO, *VAPA/B* dKO (*VAP* dKO), *ORP10* KO and *RAB7A/B* dKO (*RAB7* KO) THP-1 cell lines, two sgRNA sequences for each gene were designed using the website https://www.benchling.com. A total of $1.0 \times 10^6$ THP-1 cells were transfected with sgRNA-expressing plasmids (0.5 µg each) using X-tremeGENE 9 DNA Transfection Reagent (Roche, 6365779001) according to the manufacturer's manual. Twenty-four hours after transfection, GFP-positive and/or RFP-positive cells were enriched by fluorescence-activated cell sorting (FACS) (BD FACSAria II, BD Biosciences). Single cell colonies were obtained by seeding them into 96-well plates by series of dilution. Obtained single-cell KO clones were validated by immunoblotting or Sanger sequencing of the PCR-amplified targeted fragment after cloning into the pUC57 vector. The following primers were used for PCR amplification of the SYS1 targeted fragment: 5′-CGAATGCATCTAGA TATCGGATCCACTCTGAGAATGGGTCTGTCTGCCC-3′ and 5′-GCCTCT GCAGTCGACGGGCCCGGGTTCCCAGGGCTACAAAGAAAGAAGG-3′.

To generate *Arfrp1* KO and *Sys1* KO iBMDM cell lines, we designed two sgRNAs for each gene and cloned them into lentiCRISPR v2. Lentiviruses were produced as described above. Twenty-four hours after lentiviral infection, sgRNA-expressing iBMDMs were selected by treatment with 2 µg ml⁻¹ puromycin for 24 h. Single cell colonies were obtained by seeding them into 96-well plates by series of dilution. Obtained *Arfrp1* KO and *Sys1* KO single-cell clones were validated by immunoblotting or Sanger sequencing.

To generate the *OSBP*, *PI4KIIα* or *PI4KIIβ* KO THP-1 cell pool, we designed two sets of sgRNAs (two sgRNAs for each gene in each set) and cloned them into lentiCRISPR v2 puro or bsd. Lentiviruses were produced as described above. THP-1 cells were infected with sgRNA-expressing lentivirus as indicated. Forty-eight hours after lentiviral infection, sgRNA-expressing cells were selected by treatment with 2 µg ml⁻¹ puromycin or 30 µg ml⁻¹ blasticidin S for 24 h. After two splitting, cells were used for further experiments.

## Immunoblotting

After treatments, cell culture supernatants and cell lysates were collected for immunoblotting analysis. Cell lysates were collected in 1× SDS sample buffer or in 1× RIPA buffer (50 mM Tris-HCl (pH 7.5), 150 mM NaCl, 1% Triton X-100, 1 mM EDTA, 1 mM EGTA, 2 mM sodium pyrophosphate, 1 mM NaVO₄ and 1 mM NaF) supplemented with a protease inhibitor cocktail. The immunoblots were prepared using Tris-glycine SDS−PAGE. For the supernatants, the proteins were extracted using methanol−chloroform precipitation, separated by tricine SDS−PAGE and analyzed by immunoblotting. The immunoblots were probed overnight at 4 °C or 1 h at room temperature 22–23 °C with primary antibody. After three washes with TBST, membranes were incubated with HRP-conjugated secondary antibody for 1 h at room temperature. Membranes were again washed with TBST three times and were incubated with Immobilon Forte Western HRP substrate (EMD Millipore, WBLUF0500). Images were captured using an AI600 imager (GE Healthcare Life Sciences). The primary and secondary antibodies used for immunoblotting in this study are listed in in Supplementary Table 2.

## Immunofluorescence microscopy

Immunofluorescence analyses in Figs. 1a, 1b, 3a, 3b, 4c–e, 6e and 6f; Extended Data Figs. 1a–j, 3c, 3d, 4a–c, 5a, 5b, 6a and 6b; and Supplementary Fig. 3 were performed as previously described[12]. Cells were imaged using a Plan-Apochromat 63× or 100×/1.4 oil objective on a Zeiss LSM800 or LSM880 confocal system equipped with an Airyscan module and controlled by the Zen blue software. For the rest, cells plated on coverslips (9–15 mm) were fixed in 4% paraformaldehyde

for 15 min at room temperature after the indicated treatments. Cells were permeabilized in phosphate-buffered saline (PBS) containing 0.1% saponin for 10 min. After blocking with 10% normal goat serum in PBS for 1 h, cells were incubated with primary antibodies for 1 h at room temperature. After incubation with secondary antibodies for 1 h at room temperature, cells were stained with 4,6-diamidino-2-phenylindole (DAPI) and mounted with ProLong Gold Antifade Mountant (Thermo Fisher Scientific, P36934). Intracellular PI4P staining was performed as previously described in ref. [36]. Images were acquired below pixel saturation using a Leica TCS SP8 confocal laser scanning microscope (Leica Microsystems). Fluorescence micrographs are representative of images collected from at least three independent experiments. The images used for phenotype quantification were acquired with the same parameters (that is, digital gain, laser power and magnification) and processed with Fiji software (ImageJ; National Institutes of Health, NIH). Brightness and contrast were adjusted with Adobe Photoshop, and figure panels were assembled with Adobe Illustrator.

Antibodies used for immunofluorescence analysis in this study are listed in Supplementary Table 2.

All of the antibodies used in this study have been experimentally validated. The fluorescence-minus-one (FMO) controls for all fluorescently labeled antibodies are provided in Supplementary Fig. 4.

## Live-video imaging

For the live-video imaging of actin comets, HeLa cells were plated in glass-bottomed dishes (MatTek) and co-transfected with SNX2-eGFP or NLRP3-eGFP and LifeAct-RFP. Cells were treated with vehicle or 15 µM nigericin for 60 min in the presence of 5 µg ml⁻¹ Alexa-633-conjugated transferrin and imaged with the Zeiss LSM800 microscope equipped with 488 nm, 561 nm and 647 nm argon lasers using the Plan-Apochromat NA 63×/1.4 DIC oil immersion objective. During imaging, cells were maintained in complete culture medium in a humidified atmosphere at 37 °C. Fluorescence images presented are representative of cells imaged in at least three independent experiments and were processed with Fiji software. One hundred frames were taken with 6 s intervals for each condition.

## Uptake and trafficking of CI-MPR antibody and Cy3-labeled StxB

CI-MPR antibody uptake was performed as previously described[37]. Cells were treated with 10 µM nigericin or 45 µg ml⁻¹ CL097 in serum-free medium. At 5 min after nigericin addition or 20 min after CL097 addition, an anti-CI-MPR antibody (10 µg ml⁻¹; Thermo Fisher Scientific, MA1-066) was added into culture medium. Cells were fixed at 40 min after the incubation of CI-MPR antibody and co-stained with antibodies against GCC1 or EEA1. For StxB uptake, Cy3-conjugated StxB was added into culture medium at 5 min after nigericin (10 µM) addition or 20 min after CL097 (45 µg ml⁻¹) addition. Cells were fixed at 40 min after the incubation of Cy3-conjugated StxB and stained with cognate antibodies. Images were acquired using the Leica TCS SP8.

## Detection and quantification of intracellular PI4P levels

Intracellular PI4P levels were assessed using two independent approaches: indirect immunofluorescence using an antibody against PI4P (Echelon Biosciences, Z-P004), and fluorescence of a GFP-tagged genetically encoded PI4P probe, SidC[38]. Both of these approaches provided highly matched intracellular PI4P distribution patterns shown in Supplementary Data 1. For the indirect immunofluorescence, cells were stained with the antibody against PI4P following the protocol described in ref. [36]. Images from 10–15 fields (each containing 5–10 cells) were randomly taken below pixel saturation with the same microscope settings (that is, laser power and detector amplification). The mean intensity per cell was determined using Fiji software. For the assessment of PI4P levels, a mask using EEA1 or golgin-97 was generated for each cell. The mean intensities of both PI4P and EEA1 or Golgin97 were

measured in these regions. After background subtraction, the PI4P values were normalized singularly by the intensity of EEA1 or Golgin97. The relative values are expressed by further normalization against the control condition. To analyze the peripheral number of PI4P-positive spots, we used the 'Analyze Particles' tool in Fiji after subtraction of the PI4P signal in the Golgi area. For the fluorescence of GFP-SidC, HeLa cells expressing SidC-GFP were fixed after treatment as indicated. Cells were stained with EEA1, Golgin97 or TGN46. Golgin97 signal was used to generate a mask for the Golgi region. The mean intensity of GFP-SidC was measured in the Golgi region and normalized for the mean intensity of an equivalent area outside the Golgi region. Golgi PI4P was expressed as the ratio between the intensity of GFP-SidC in the Golgi region and in the equivalent extra-Golgi area.

### Quantitative measurement of membrane contact sites

**FRET/FLIM-based reporter system.** FRET/FLIM analysis of RAB5-GFP alone and in combination with mCherry-Cb5 was performed as previously described[12]. Fluorescence-lifetime imaging microscopy (FLIM) data analysis was performed using SymPhoTime 64 (PicoQuant). In brief, for RAB5-GFP detection, after fixation, samples were excited using a pulsed laser (femtosecond Ti:Sa laser, Chameleon Vision 2, Coherent; set at 900 nm) at a repetition rate of 80 MHz. The fluorescence signal was spectrally filtered using a narrowband emission filter (BP filter 482/35; Chroma). Photons were temporally collected using a single-photon sensitive detector (PMA Hybrid 40, PicoQuant) combined with a single photon counting module (TimeHarp 260, PicoQuant). FLIM data were background-subtracted and processed using the pixel-based fitting software SymPhoTime 64 to calculate the lifetime maps. Lifetime was measured for each single thresholded pixel. Amplitude-weighted average lifetime was calculated as $\tau = \Sigma(\alpha_i \tau_i)/\Sigma\alpha_i$, where $\alpha_i$ is the amplitude of each lifetime $\tau_i$. FRET/FLIM efficiency was calculated by applying the following equation: FRET efficiency $= 1 - (\tau_{DA}/\tau_D)$, where $\tau_{DA}$ is the lifetime of the donor in the presence of the acceptor and $\tau_D$ is the lifetime of the donor without the acceptor[39].

**Rapamycin-induced reporter system.** In cells expressing FRB and FKBP domain-fused membrane-localized reporters, short treatment with low concentrations of rapamycin stabilizes pre-existing physiological membrane contact sites without inducing artificial tethering[12]. To measure EECS, iRFP-FKB-Rab5 and mCherry-FKBP-Cb5 were expressed in HeLa cells; to measure ER–TGN contact sites, TGN46-FRB-HA-GFP-T2A-mCherry-FKBP-Cb5 was expressed in HeLa cells. Cells were further treated with vehicle, nigericin or CL097 for the indicated time points followed by incubation with or without 200 nM rapamycin for 4 min before paraformaldehyde fixation. Images from 15–20 individual fields were captured (for a total of 100–120 cells for each group) to assess the localization of mCherry-Cb5. P values were calculated on the basis of mean values from four independent experiments.

### Analysis of Sytox Green uptake using flow cytometry

After treatment, cells were detached with 5 mM EDTA and incubated with Sytox Green (Thermo Fisher Scientific, S7020) at 1/8,000 dilution. Cells were then analyzed using a BD FACSCelesta cell analyzer (BD Biosciences). The percentage of Sytox Green-positive cells was analyzed using FlowJo v10.8.1 software. The gating strategy is shown in Supplementary Fig. 5.

### LPS-induced endotoxemia in mice

$Arfrp1^{fl/fl}$ and $Arfrp1^{fl/fl};LysM-Cre$ mice at 6–8 weeks of age were used. Mice were peritoneally injected with 15 mg kg$^{-1}$ LPS. Vehicle or 50 mg kg$^{-1}$ MCC950 was peritoneally injected at 1 h before LPS injection. For the analysis of serum cytokines, blood was collected at 3 h after LPS injection. For survival curves, 2.5 mg kg$^{-1}$ buprenorphine was peritoneally administered every 6 h, and the survival of mice was monitored every 6 h.

### Measurement of cytokines using ELISA

Human IL-1β, human TNF-α, mouse IL-1β and mouse TNF-α in cell culture supernatants or serum were measured using Human IL-1β/IL-1F2 DuoSet ELISA (R&D Systems, DY201), Human TNF-α DuoSet ELISA (R&D Systems, DY210), Mouse IL-1β/IL-1F2 DuoSet ELISA (R&D Systems, DY401) and Mouse TNF-α DuoSet ELISA (R&D Systems, DY410), separately, according to the manufacturer's protocols.

### Image quantification

Image processing and quantification were performed using Fiji software. For the quantification of StxB and CI-MPR on endosomes and Golgi, to define regions of interest (ROIs), we thresholded the EEA1 (for endosomes) and GCC1 (for Golgi) channels. Defined ROIs was superimposed onto the channel of interest (StxB and CI-MPR). Fluorescence in the defined ROIs were measured by recording 'RawIntDen' (the sum of the values of the pixels in the selection). The percentages of StxB and CI-MPR in defined ROIs were calculated by dividing RawIntDen in the defined ROIs by RawIntDen of the whole cell. A single mid-Z-stack image was analyzed per cell. Co-localization analysis was carried out by subtracting background and measuring the Pearson's correlation coefficient using the Fiji co-localization plug-in 'Coloc2'. The percentages of cells with NLRP3 puncta, peripheral PI4P spots and TGN46 dispersal were analyzed using the 'Analyze Particles' tool in Fiji to determine their number in each cell. For each experiment, images were acquired below pixel saturation, and the same threshold was chosen and applied.

### Statistical analyses

Preliminary experiments were performed and sample size was determined based on generally accepted rules to test preliminary conclusions reaching statistical significance, where applicable. Unless specified, statistical analyses were performed with the unpaired two-sided t-test using GraphPad Prism. For serum cytokine levels, statistical analyses were performed with the Mann–Whitney U-test. For survival curves, statistical analysis was performed with the log-rank (Mantel–Cox) test.

### Reporting summary

Further information on research design is available in the Nature Portfolio Reporting Summary linked to this article.

## Data availability

All data are available in the main text, extended data figures or supplementary materials. Source data are provided with this paper.

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

## Acknowledgements

We acknowledge Anne Spang at the Biozentrum, University of Basel, for providing HeLa cells stably expressing mApple-tagged RAB5. We thank Fabien Alpy and Izabela Sumara for helpful suggestions and discussions. We also thank the scientific platforms at the IGBMC, including the cell cytometry facility, the cell culture facility, the imaging facility and the animal facility, for their help during the entire project. Work in the laboratory of R.R. was supported by Agence nationale de la recherche (ANR) (AAPG 2017 LYSODIABETES), the USIAS fellowship grant 2017 of the University of Strasbourg, the Fondation pour la recherche médicale (FRM) – Program: Equipe FRM (EQU201903007859, Prix Roger PROPICE pour la recherche sur le cancer du pancréas), grants from French State funds managed by the ANR under the frame program Investissements d'Avenir (ANR-10-LABX-0030-INRT and ANR-11-INBS-0009-INGESTEM). L.R. acknowledges the support from the China Scholarship Council (CSC). M.A.D.M. acknowledges the support from the European Research Council Advanced Investigator grant 670881 (SYSMET), the Italian Association for Cancer Research (grant IG 2017_20815), and the Italian Ministry of University and Research (PRIN, 2020PKLEPN). R.V. acknowledges the support from the Italian Association for Cancer Research (grant MFAG 2020_25174). A.S. acknowledges the support from the 'German Research Foundation [DFG, SFB 958]'. J.S.B. acknowledges the support from the Intramural Program of the Eunice Kennedy Shriver National Institute of Child Health and Human Development (NICHD) and the NIH (project ZIA HD001607 to J.S.B.).

## Author contributions

Conceptualization, Z.Z., R.R., M.A.D.M. and J.S.B.; methodology, Z.Z., R.V., L.R. and Z.L.; investigation, Z.Z., R.V., L.R., K.V. and Z.L.; writing (original draft), Z.Z.; writing (review and editing), R.R., Z.Z., J.S.B., A.S. and M.A.D.M.; funding acquisition, R.R. and M.A.D.M.; resources, Z.Z., R.V., J.S.B. and A.S.; supervision, R.R., Z.Z. and M.A.D.M.

## Competing interests

The authors declare no competing interests.

## Additional information

**Extended data** is available for this paper at https://doi.org/10.1038/s41590-022-01355-3.

**Correspondence and requests for materials** should be addressed to Zhirong Zhang, Maria Antonietta De Matteis or Romeo Ricci.

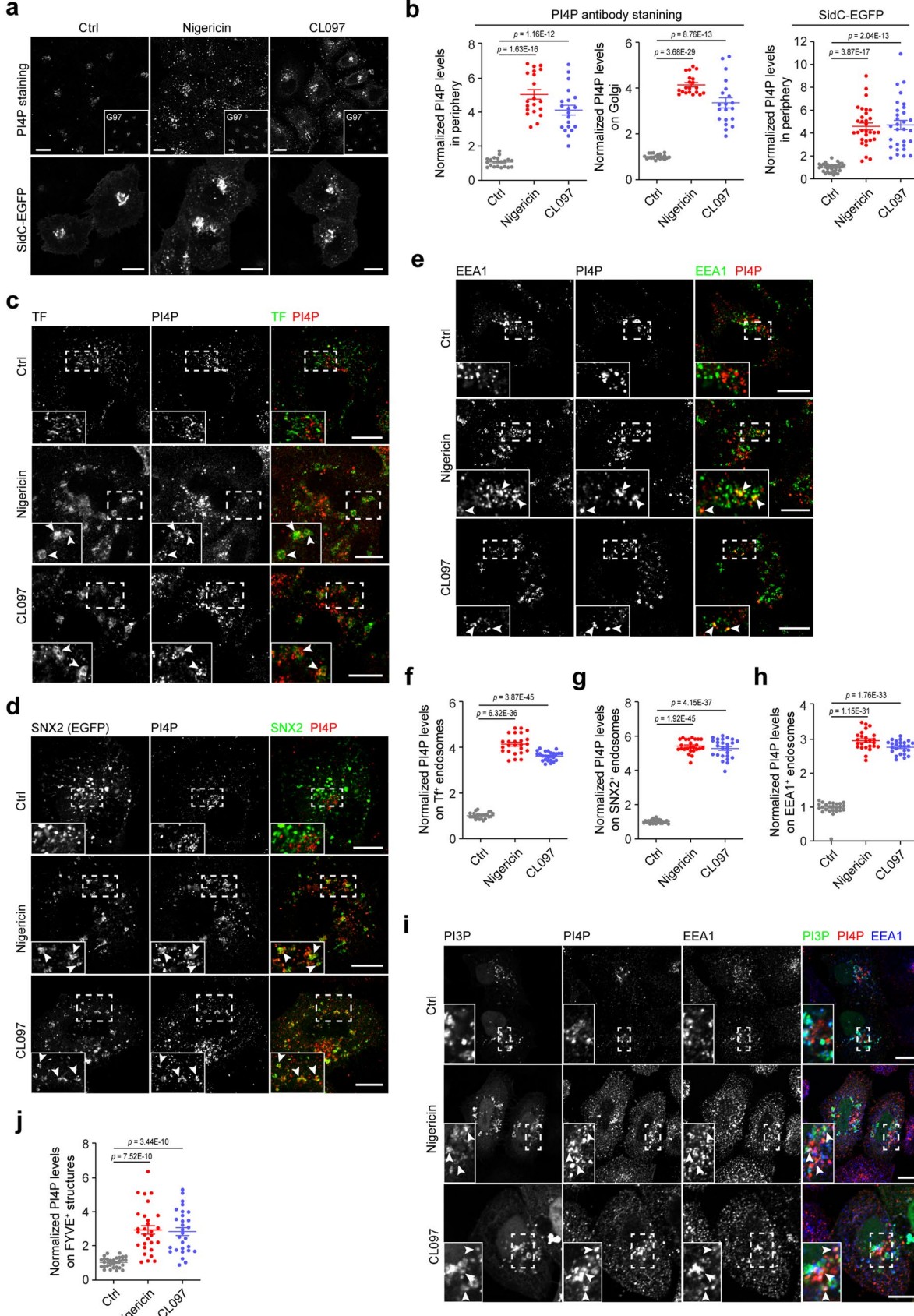

**Extended Data Fig. 1 | See next page for caption.**

**Extended Data Fig. 1 | NLRP3 activators trigger PI4P accumulation on endosomes in HeLa cells. a**, Confocal images of WT and SidC-GFP-expressing HeLa cells treated with vehicle (Ctrl), 15 µM nigericin or 45 µg ml$^{-1}$ CL097 for 60 min. An antibody against PI4P and against the TGN marker Golgin97 (G97) was used (lower left corner) or the distribution of SidC-GFP was analyzed. Scale bar: 10 µm. **b**, Quantification of PI4P levels in cells in experiments shown in panel **a**. N = 3, n = 20 cells for each group. **c**, Airyscan images of HeLa cells treated with vehicle (Ctrl), 15 µM nigericin or 45 µg ml$^{-1}$ CL097 and 5 µg ml$^{-1}$ Alexa488-conjugated Transferrin (Tf) for 60 min. After fixation, cells were stained with an antibody against PI4P. Magnifications of areas in dashed squares shown in the lower left corner. Arrowheads indicate TF-positive structures containing PI4P. Scale bar: 10 µm. **d**, Airyscan images of HeLa cells expressing SNX2-eGFP treated as in experiment shown in panel **a**. Cells were stained with an antibody against PI4P. Magnifications of areas in dashed squares shown in the lower left corner. Arrowheads indicate SNX2-eGFP-positive structures containing PI4P. Scale bar:

10 µm. **e**, Airyscan images of HeLa cells treated as in experiment shown in panel **a**. Cells were co-stained with antibodies against PI4P and EEA1. Magnifications of areas in dashed squares shown in the lower left corner. Arrowheads indicate EEA1-positive structures containing PI4P. Scale bar: 10 µm. **f, g, h**, Quantification of PI4P levels on Tf-, SNX2-eGFP and EEA1-positive structures in cells in experiments as described for panel **c**. Means ± SD, N = 3, n = 30 cells for each group. **i**, Confocal images of HeLa cells expressing GFP-tagged FYVE domain of SARA treated as in experiment shown in panel a. After fixation, cells were co-stained with an antibody against PI4P and EEA1. Magnifications of areas in dashed squares shown in the lower left corner. Arrowheads indicate FYVE-GFP-positive structures containing PI4P and EEA1. Scale bar: 10 µm. **j**, Quantification of PI4P levels on FYVE-GFP-positive structures in experiments shown in panel **i**. Means ± SD, N = 3, n = 30 cells for each group. Data were analyzed with an unpaired two-sided *t*-test (**b, f, g, h**). Data shown in **a, c, d, e, i** are representative of at least three independent experiments.

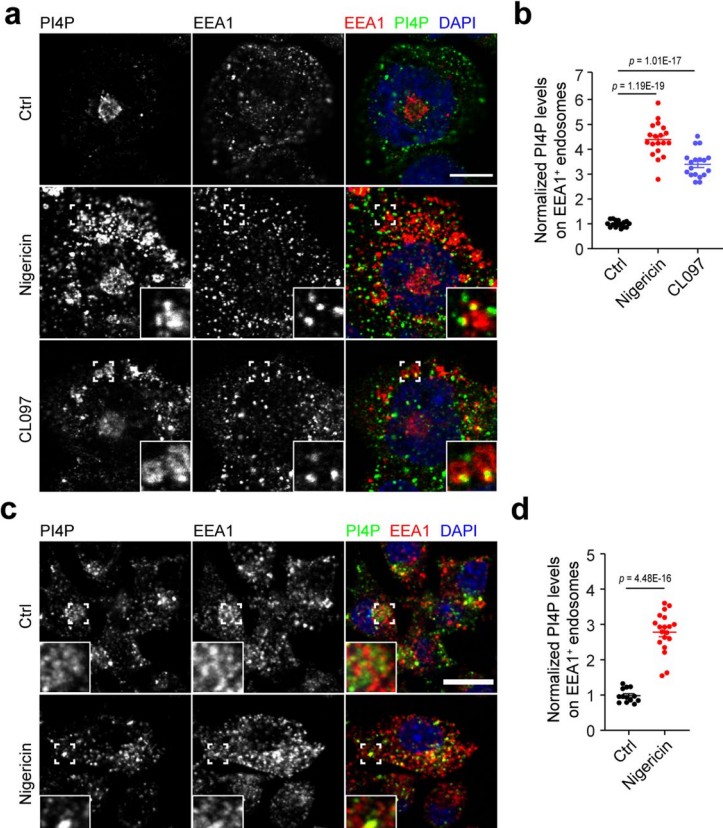

**Extended Data Fig. 2 | NLRP3 activators trigger PI4P accumulation on endosomes in THP-1 cells and iBMDMs. a**, Confocal images of *ASC* KO THP-1 cells treated with vehicle (Ctrl), 15 μM nigericin or 50 μM CL097 for 30 min. After fixation, cells were co-stained with antibodies against PI4P and EEA1. DAPI was used to stain the nucleus. Magnifications of areas in dashed squares shown in the lower right corner. Scale bar: 10 μm. **b**, Quantification of PI4P levels on EEA1-positive structures in experiments shon in panel **a**. Means ± SD, N = 3, n = 16 cells for control-treated group; n = 19 for nigericin-treated group and n = 18 cells for CL097-treated group. **c**, Confocal images of *ASC* KO iBMDMs treated with vehicle or 15 μM nigericin for 30 min. Cells were co-stained with antibodies against PI4P and EEA1. Magnifications of areas in dashed squares shown in the lower left corner. Scale bar: 10 μm. **d**, Quantification of PI4P levels on EEA1-positive structures in experiments shown in panel **c**. Means ± SD, N = 3, n = 13 cells for control-treated group and n = 19 cells for nigericin-treated group. Data were analyzed with an unpaired two-sided *t*-test (**b**, **d**). Data shown in **a**, **c** are representative of at least three independent experiments.

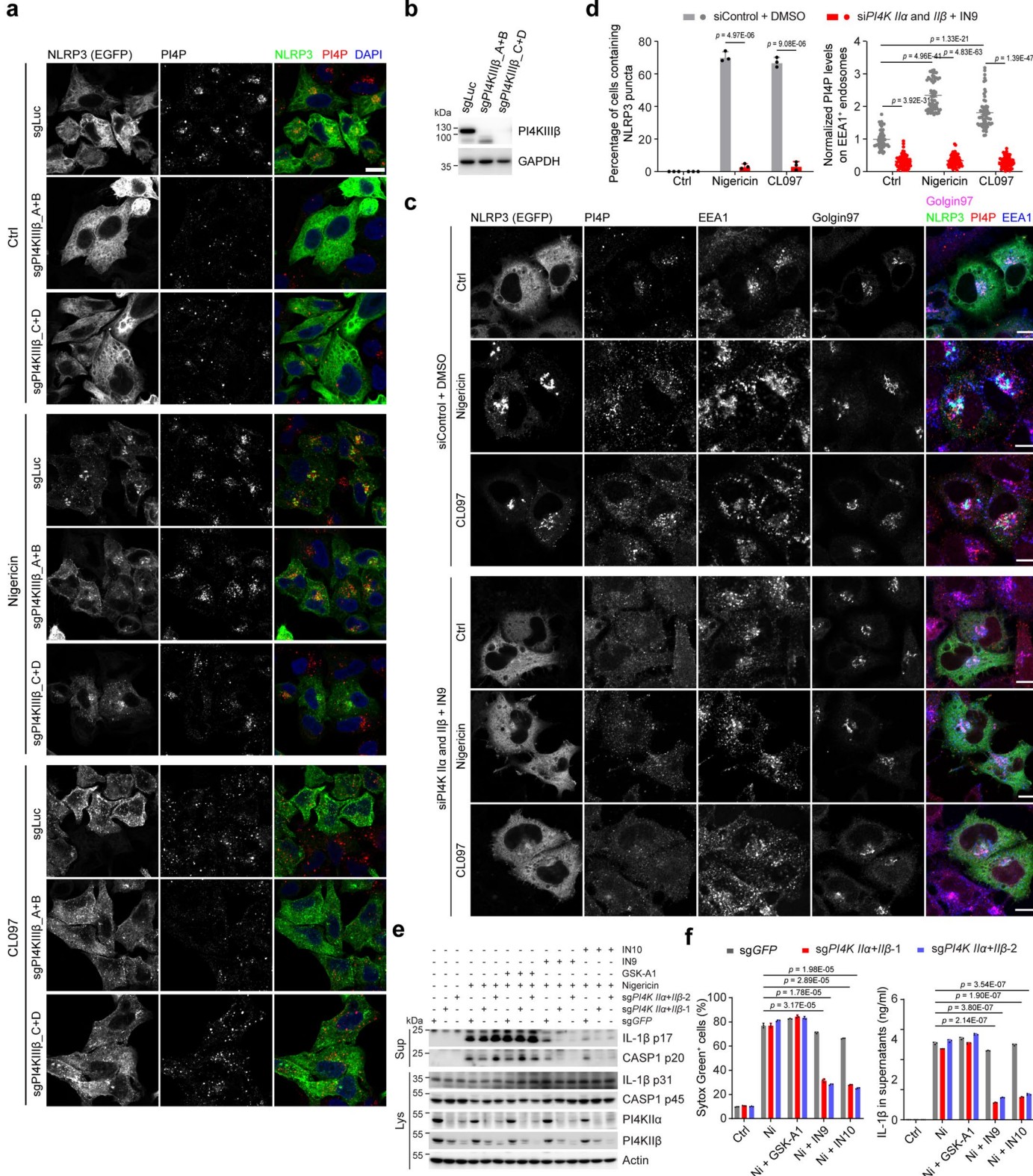

**Extended Data Fig. 3 | See next page for caption.**

**Extended Data Fig. 3 | Overlapping functions of PI4KIIα, PI4KIIβ and PI4KIIIβ in promoting endosomal PI4P accumulation and NLRP3 inflammasome activation. a**, Confocal images of HeLa cells stably expressing eGFP-tagged NLRP3 co-expressing sgRNAs targeting *Luciferase* (sg*Luc*) or *PI4KIIIβ* (sgPI4K*IIIβ*). Two different pairs of sgRNAs were used to delete *PI4KIIIβ* (pairs A + B and C + D as indicated in the methodology). Cells were treated with vehicle, 15 μM nigericin for 30 min or 45 μg ml$^{-1}$ CL097 for 80 min and then stained with an antibody against PI4P. DAPI was used to stain the nucleus. Scale bar: 10 μm. **b**, Immunoblotting of lysates from HeLa cells used in experiments shown in panel **a**. **c**, Confocal images of NLRP3-eGFP-expressing HeLa cells treated with control siRNA or PI4KIIα and PI4KIIβ siRNA. After 72 h siRNA treatment, cells were treated or not with 15 μM nigericin or 45 μg ml$^{-1}$ CL097 for 60 min in presence of DMSO or 50 nM of the PI4KIIIβ inhibitor IN9. Cells were co-stained with antibodies against PI4P, Golgin97 and EEA1. Scale bar: 10 μm. **d**, Quantification of PI4P levels on EEA1-positive endosomes in cells (right, Means ± SD, N = 3, n = 90 cells for each group) and cells containing NLRP3 puncta (left, means of percentage ± SD, N = 3, n = 100 cells for each group) in experiments shown in panel **c**. **e**, Immunoblotting of supernatants and lysates from THP-1 cells expressing sgRNAs targeting *GFP* (sg*GFP*) or *PI4KIIα* and *PI4KIIβ* (sg*PI4K IIα + IIβ*). After pretreatment with DMSO, the PI4KIIIα inhibitor GSK-A1 (100 nM) or the PI4K IIIβ inhibitors IN9 (100 nM) and IN10 (100 nM) for 30 min, cells were treated with 15 μM nigericin in presence of DMSO or inhibitors as indicated for 45 min. Antibodies against proteins as indicated were used. Actin was chosen as a loading control. **f**, Cellular uptake of Sytox Green and ELISA analysis of IL-1β secretion in experiments shown in panel **e**. Means ± SD, N = 3. Data were analyzed with an unpaired two-sided *t*-test (**d**, **f**). Data shown in **a**, **b**, **e** are representative of at least three independent experiments.

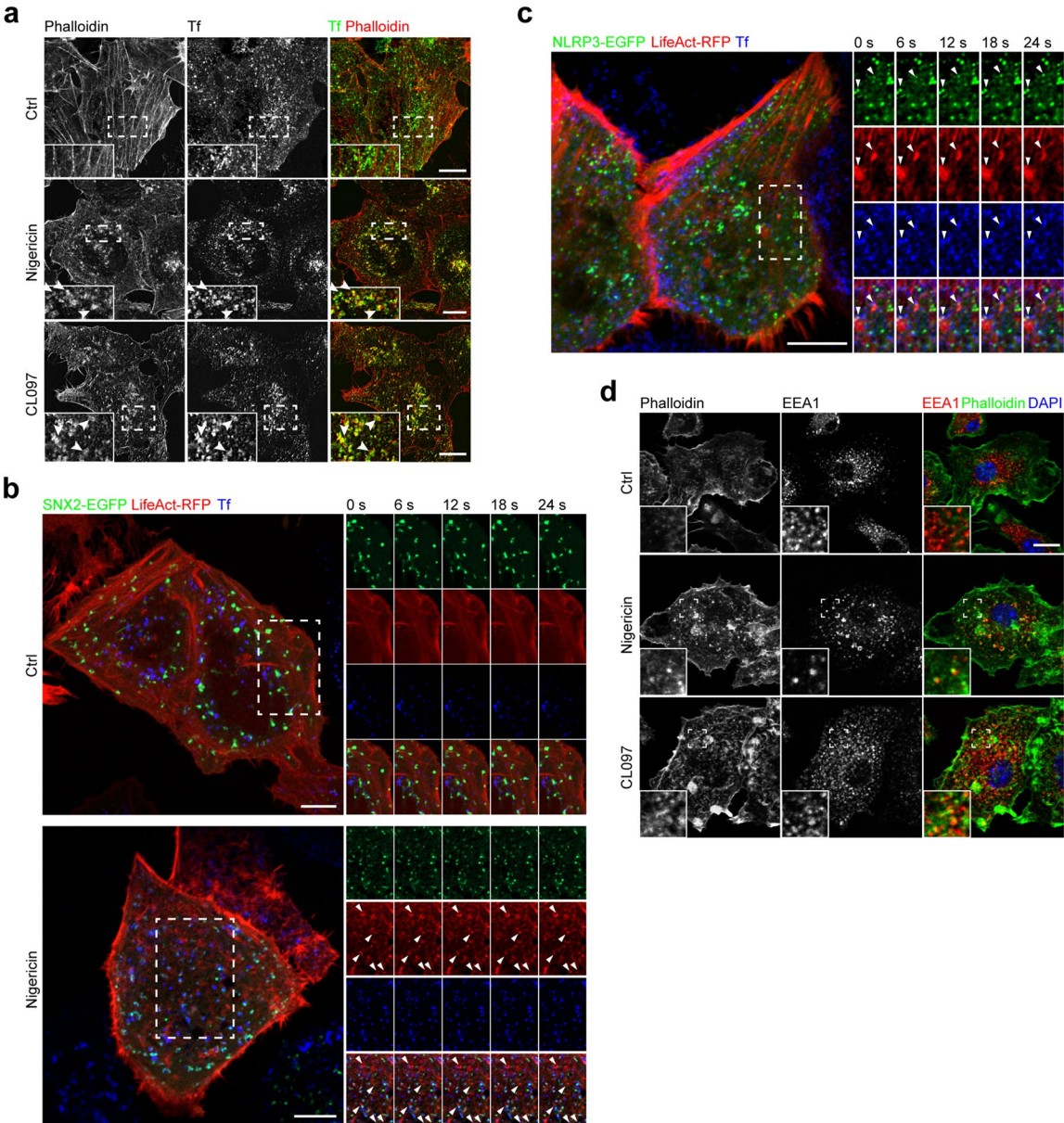

**Extended Data Fig. 4 | NLRP3 activators disrupt EECS inducing actin comets propelling early endosomes. a**, Airyscan images of HeLa cells loaded with 5 μg ml⁻¹ Alexa633-conjugated Transferrin (Tf). Cells were treated with vehicle (Ctrl), 15 μM nigericin or 45 μg ml⁻¹ CL097 for 60 min. Cells were stained with Alexa546-conjugated Phalloidin. Magnifications of areas in dashed squares shown in the lower left corner. Arrowheads indicate co-localization of phalloidin and Tf. Scale bar: 10 μm. **b**, Live-video imaging of HeLa cells expressing LifeAct-RFP and SNX2-eGFP treated with control (top) or 15 μM nigericin for 60 min. Cells were imaged with 6 second intervals for about 10 min. 5 μg ml⁻¹ Alexa633-conjugated Transferrin (Tf) was added to visualize the endosomal compartment. ROIs are indicated with dashed squares and 5 frames of ROIs with individual channels and merged images are shown. Arrowheads indicate persisting polymerized actin.

Scale bar: 10 μm. **c**, Live-video imaging of HeLa cells expressing both LifeAct-RFP and NLRP3-eGFP treated with 15 μM nigericin for 60 min. Cells were imaged with 6 second intervals for about 10 min. 5 μg ml⁻¹ Alexa633-conjugated Transferrin (Tf) was added to visualize the endosomal compartment. ROI is indicated with dashed squares and 5 frames of ROI with individual channels and merged images are shown. Arrows indicate the formation of NLRP3 puncta on TF-loaded endosomes propelled by actin comets. Scale bar: 10 μm. **d**, Confocal images of *NLRP3* KO BMDMs treated with vehicle (Ctrl), 7.5 μM nigericin or 50 μM CL097 for 20 min. After fixation, cells were co-stained with Alexa488-Phalloidin and an antibody against EEA1. DAPI was used to stain the nucleus. Magnifications of areas in dashed squares shown in the lower left corner. Scale bar: 10 μm. Data shown are representative of at least three independent experiments.

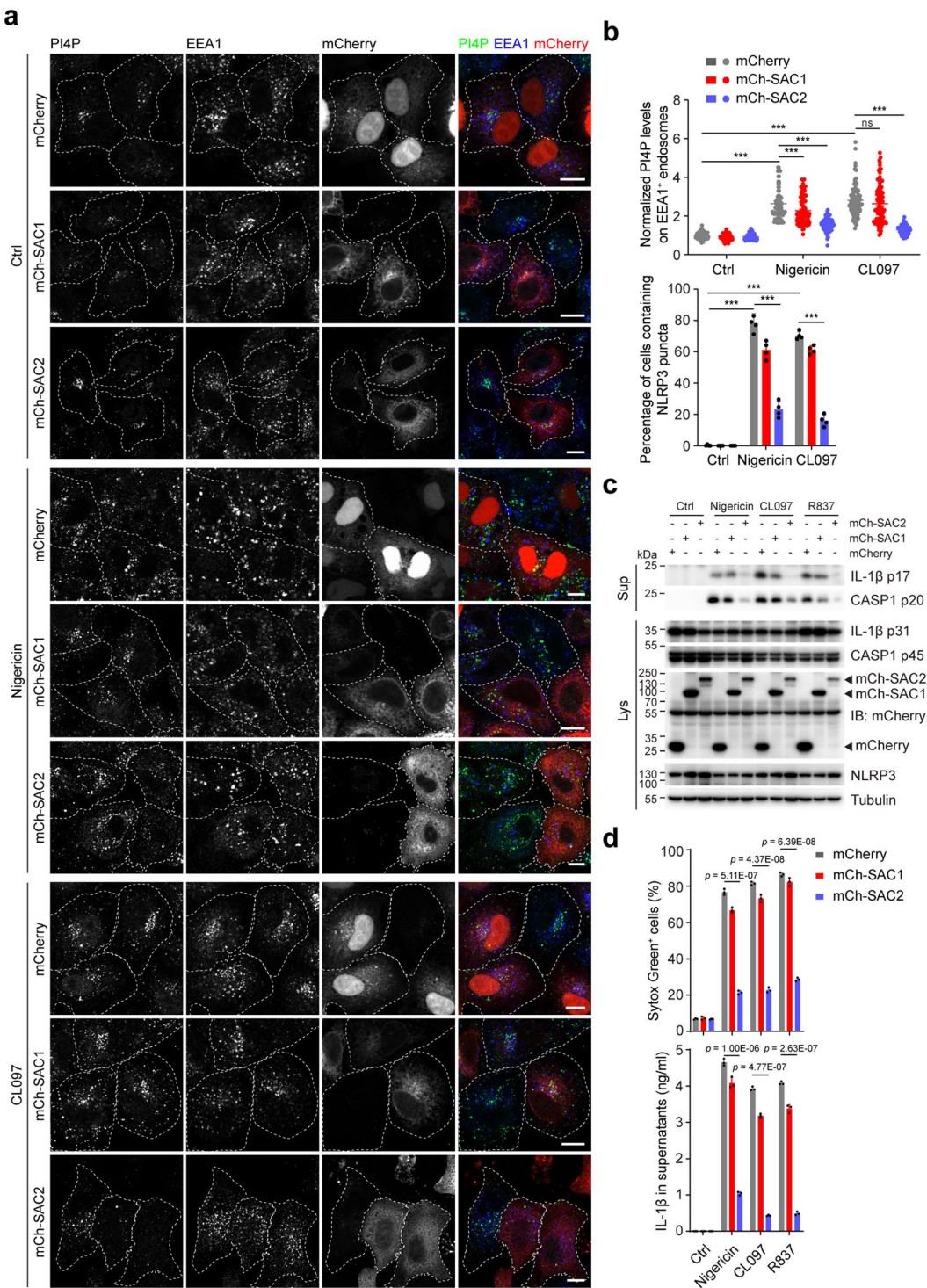

**Extended Data Fig. 5 | Overexpression of SAC2, but not SAC1 prevents endosomal PI4P accumulation, endosomal recruitment of NLRP3 and inflammasome activation. a**, Confocal images of HeLa cells expressing mCherry, mCherry-tagged hSAC1 (mCh-SAC1) or mCherry-tagged hSAC2 (mCh-SAC2) treated with vehicle (Ctrl), 15 μM nigericin or 45 μg ml⁻¹ CL097 for 60 min. Cells were co-stained with antibodies against PI4P and EEA1. Scale bar: 10 μm. **b**, Quantification of PI4P levels on endosomes in HeLa cells (upper panel, Means ± SD, N = 3, n = 90 cells for each group) and NLRP3 puncta-containing HeLa cells (lower panel, means of percentage ± SD, N = 4, n = 100 cells for each group) in experiments shown in panel **a**. **c**, Immunoblotting of supernatants and lysates from THP-1 cells stably expressing mCherry, mCherry-tagged hSAC1 (mCh-SAC1)

or mCherry-tagged hSAC2 (mCh-SAC2). Cells were treated with vehicle (Ctrl), 10 μM nigericin for 45 min, 50 μM CL097 for 45 min or 100 μM R837 for 2 hours. Antibodies recognizing both p45 and p20 fragments of caspase-1 (CASP1), p31 and p17 fragments of IL-1β, NLRP3 and mCherry were used. Tubulin was chosen as a loading control. **d**, Cellular uptake of Sytox Green and ELISA analysis of IL-1β secretion of THP-1 cells stably expressing mCherry, mCherry-tagged hSAC1 (mCh-SAC1) or mCherry-tagged hSAC2 (mCh-SAC2). Cells were treated as described for panel **c**. Means ± SD, N = 3. Data were analyzed with an unpaired two-sided *t*-test (**b**, **d**). Data shown in **a**, **c** are representative of three independent experiments.

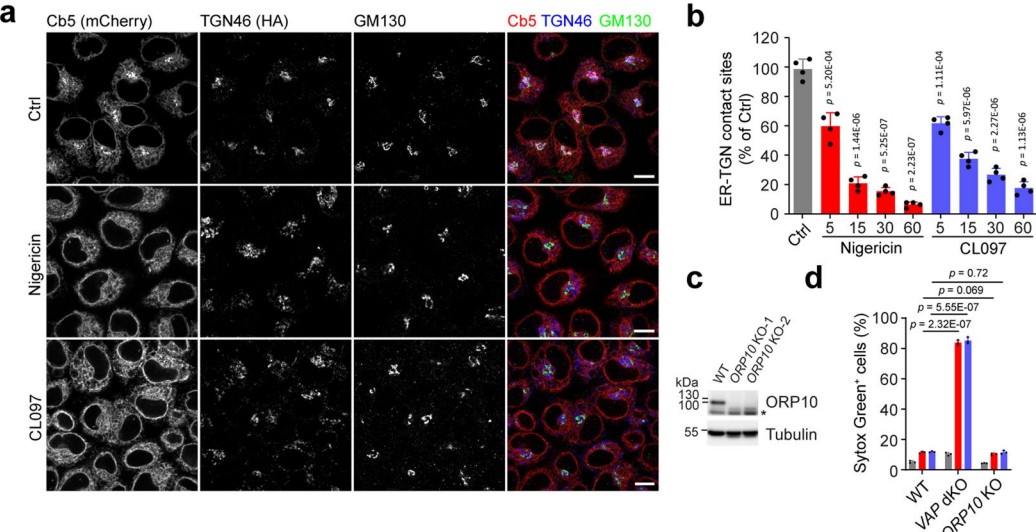

**Extended Data Fig. 6 | Disruption of ER-TGN contact sites does not contribute to NLRP3 inflammasome activation. a**, Confocal images of HeLa cells expressing the ER-TGN membrane contact site reporters mCherry-FKBP-Cb5-mCherry and TGN46-FRB-HA treated with vehicle (Ctrl), 15 μM nigericin or 45 μg ml⁻¹ CL097 for 60 min. Cells were incubated with 200 nM rapamycin for 4 min before fixation. Cells were co-stained with antibodies against HA tag and GM130. Scale bar: 10 μm. **b**, Quantification of cells showing co-localization of mCherry-FKBP-Cb5-mCherry and TGN46-FRB-HA in experiments shown in panel **a**. Cells were treated with 15 μM nigericin or 45 μg ml⁻¹ CL097 at indicated time points. Means

of percentage ± SD, N = 4, n = 100 cells for each group. **c**, Immunoblotting of lysates from WT and *ORP10* KO THP-1 cells. An antibody recognizing ORP10 was used. Tubulin was chosen as a loading control. **d**, Cellular uptake of Sytox Green in WT, *VAP* dKO and *ORP10* KO (*ORP10* KO-1) THP-1 cells treated with vehicle (Ctrl) (gray bars), 1 μg ml⁻¹ LPS (red bars) or 1 μg ml⁻¹ Pam3CSK4 (blue bars) for 2 hours. Means ± SD, N = 3. Data were analyzed with an unpaired two-sided *t*-test (**b**, **d**). Data shown in **a**, **c** are representative of three independent experiments.

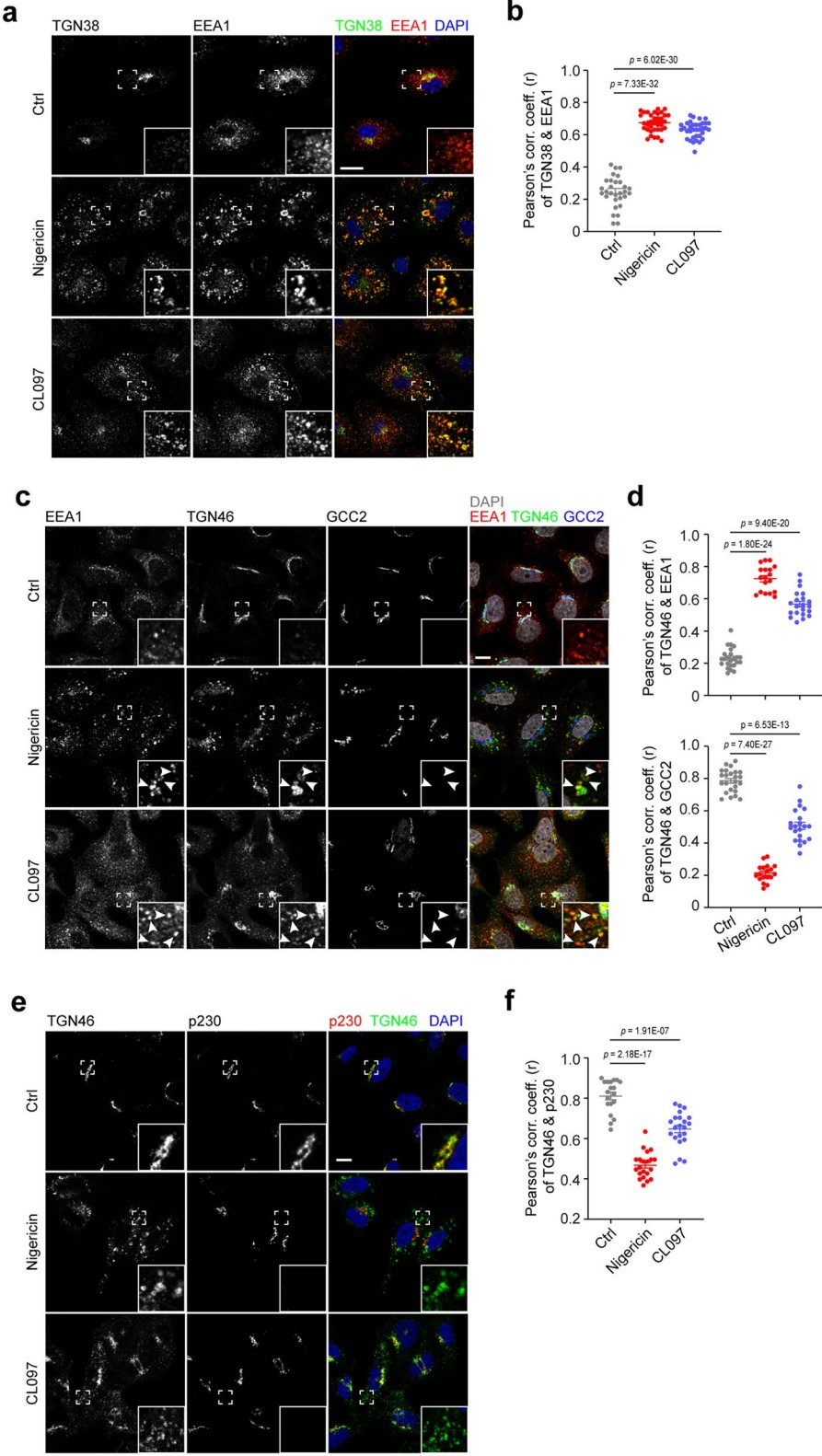

**Extended Data Fig. 7 | See next page for caption.**

**Extended Data Fig. 7 | NLRP3 activators disrupt ETT but not overall TGN integrity. a**, Confocal images of *Nlrp3* KO BMDMs treated with vehicle, 10 μM nigericin or 50 μM CL097 for 30 min. Cells were co-stained with antibodies against EEA1 and TGN38. DAPI was used to stain the nucleus. Magnifications of areas in dashed squares shown in the lower right corner. Scale bar: 10 μm. **b**, *Pearson*'s correlation coefficients of fluorescence of TGN38 and EEA1 in cells in experiments shown in panel **a**. Means ± SD, N = 3, n = 30 cells for control-treated, n = 35 cells for nigericin-treated and n = 35 cells for CL097-treated group. **c**, Confocal images of HeLa cells treated with vehicle, 10 μM nigericin for 40 min or 45 μg ml⁻¹ CL097 for 80 min. Cells were stained with antibodies against EEA1, TGN46 and GCC2. DAPI was used to stain the nucleus. Magnifications of areas in dashed squares shown in the lower right corner. Arrowheads indicate EEA1-positive endosomes containing TGN46. Scale bar: 10 μm. **d**, *Pearson*'s correlation

coefficient of fluorescence of TGN46 and EEA1 (upper) and of TGN46 and GCC2 (lower) in experiments shown in panel **c**. Means ± SD, N = 3, n = 24 cells for control-treated, n = 18 cells for nigericin-treated and n = 21 cells for CL097-treated group. **e**, Confocal images of HeLa cells treated with vehicle, 10 μM nigericin for 40 min or 45 μg ml⁻¹ CL097 for 80 min. Cells were co-stained with antibodies against TGN46 and p230. DAPI was used to stain the nucleus. Magnifications of areas in dashed squares and shown in the lower right corner. Scale bar: 10 μm. **f**, *Pearson*'s correlation coefficient of fluorescence of TGN46 and p230 in experiments shown in panel **e**. Means ± SD, N = 3, n = 19 cells for control-treated group, n = 21 cells for nigericin-treated group and n = 22 cells for CL097-treated group. Data were analyzed with an unpaired two-sided *t*-test (**b**, **d**, **f**). Data shown in **a**, **c**, **e** are representative of at least three independent experiments.

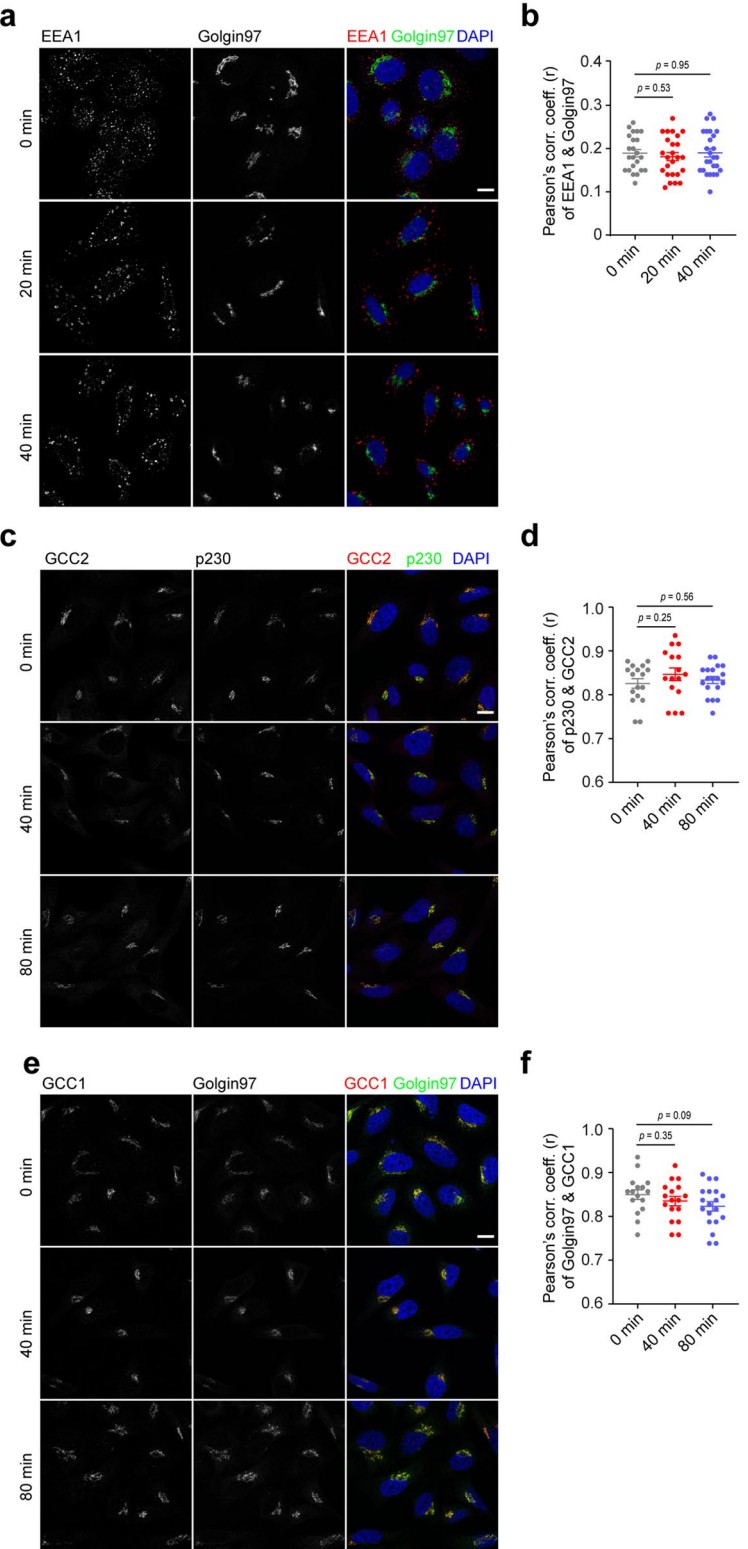

**Extended Data Fig. 8 | NLRP3 activators do not affect overall TGN integrity.**
**a**, Confocal images of HeLa cells treated with 10 μM nigericin for indicated times. Cells were co-stained with antibodies against EEA1 and Golgin97. DAPI was used to stain the nucleus. Scale bar: 10 μm. **b**, *Pearson*'s correlation coefficient of fluorescence of EEA1 and Golgin97 in experiments shown in panel **a**. Means ± SD, N = 3, n = 23 cells for '0 min', n = 25 cells for '20 min' and n = 36 cells for '40 min' group. **c**, Confocal images of HeLa cells treated with 45 μg ml⁻¹ CL097 for indicated times. Cells were co-stained with antibodies against GCC2 and p230. DAPI was used to stain the nuclear. Scale bar: 10 μm. **d**, *Pearson*'s correlation coefficient of fluorescence of GCC2 and p230 in experiments shown in panel **c**.

Means ± SD, N = 3, n = 16 cells for '0 min', n = 16 cells for '40 min' and n = 19 cells for '80 min' group. **e**, Confocal images of HeLa cells treated with 45 μg ml⁻¹ CL097 for indicated time points. Cells were co-stained with antibodies against GCC1 and Golgin97. DAPI was used to stain the nuclear. Scale bar: 10 μm. **f**, *Pearson*'s correlation coefficient of fluorescence of GCC1 and Golgin97 in experiments shown in panel **d**. Means ± SD, N = 3, n = 16 cells for '0 min', n = 17 cells for '40 min' and n = 19 cells for '80 min' group. Data were analyzed with an unpaired two-sided *t*-test (**b**, **d**, **f**). Data shown in **a**, **c**, **e** are representative of at least three independent experiments.

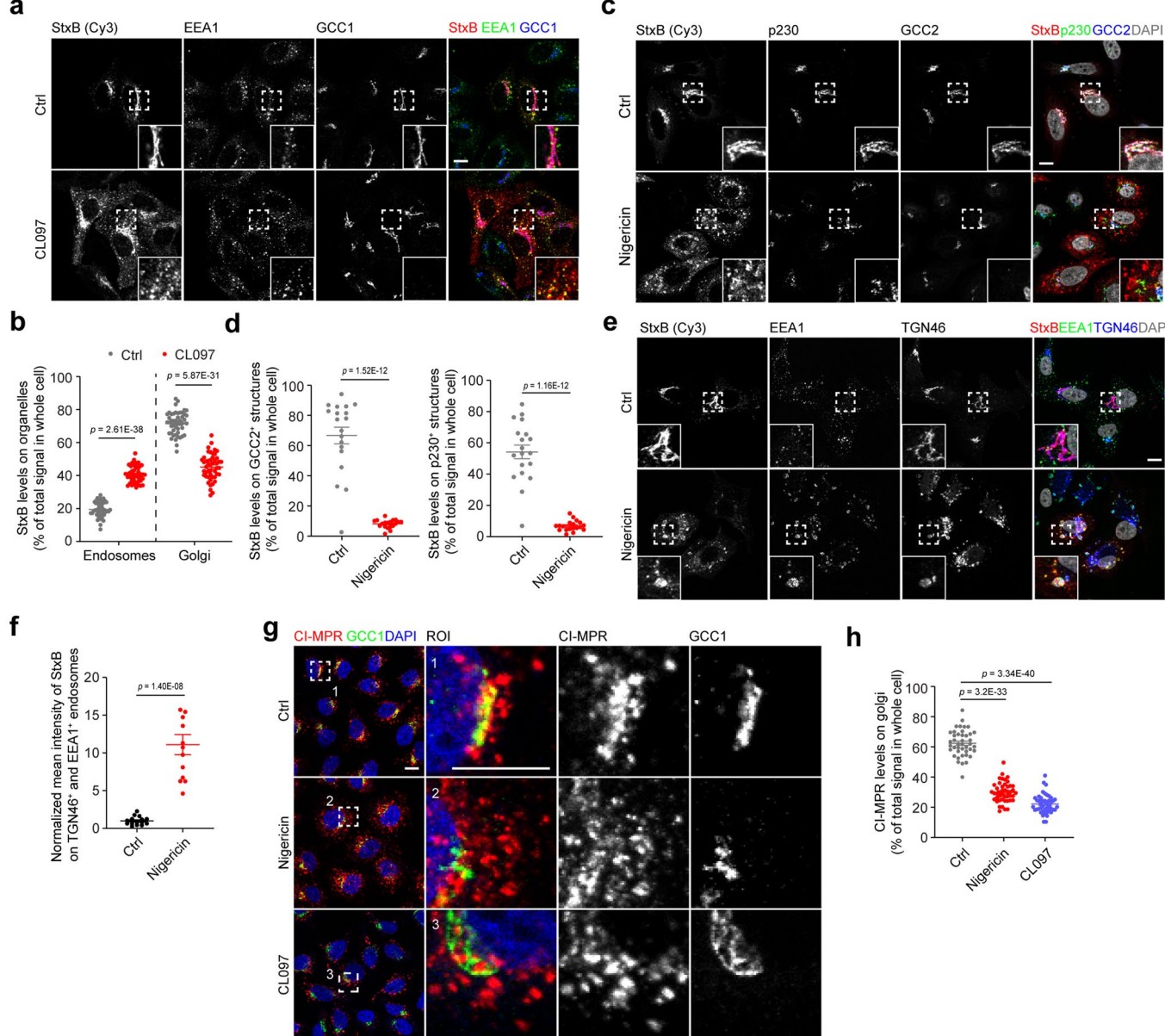

**Extended Data Fig. 9 | NLRP3 activators disrupt ETT resulting in endosomal retention of cargo. a**, StxB internalization in HeLa cells treated or not with 45 μg ml⁻¹ CL097. Cy3-conjugated StxB was added 20 min after CL097 addition. Cells were fixed 40 min after Cy3-conjugated StxB incubation and immunolabeld for EEA1 and GCC1. Magnifications of areas in dashed squares shown in the right corner. Scale bar: 10 μm. **b**, StxB level quantification on endosomes and Golgi in experiments shown in panel a. Means ± SD, N = 3, n = 47 cells for control (Ctrl)-treated and n = 48 cells for CL097-treated group. **c**, StxB internalization in HeLa cells treated or not with 10 μM nigericin. StxB labeling as described for panel **a**. Cells were immunolabeled for p230 and GCC2. Magnifications of areas in dashed squares shown in the lower right corner. Scale bar: 10 μm. **d**, StxB level quantification on p230- (right) and GCC2-positive (left) structures in experiments shown in panel **c**. Means ± SD, N = 3, n = 19 cells for control (Ctrl)-treated and n = 19 cells for nigericin-treated group. **e**, StxB internalization StxB in HeLa cells as described for panel a. Cells were immunolabeled for EEA1

and TGN46. Magnifications of areas in dashed squares shown in the lower left corner. Scale bar: 10 μm. **f**, Relative StxB intensity on EEA1-TGN46 double positive endosomes in experiments shown in panel **e**. Means ± SD, N = 3, n = 15 cells for control-treated and n = 13 cells for nigericin-treated group. **g**, CI-MPR antibody internalization in HeLa cells treated or not with 10 μM nigericin or 45 μg ml⁻¹ CL097. An antibody recognizing CI-MPR was added 5 min after nigericin or 20 min after CL097 addition. Cells were fixed 40 min after CI-MPR antibody incubation and were immunolabeled for GCC1. Magnifications of areas in numbered dashed squares shown in separate numbered images. Scale bar: 10 μm. **h**, CI-MPR level quantification on Golgi in experiments shown in panel **g**. Means ± SD, N = 3, n = 42 cells for control (Ctrl)-treated, n = 45 cells for nigericin-treated and n = 44 cells for CL097-treated group. DAPI was used to stain the nucleus in all experiments. Data were analyzed with an unpaired two-sided *t*-test (**b**, **d**, **f**, **h**). Data shown in **a**, **c**, **e**, **g** are representative of at least three independent experiments.

# Reporting Summary

## Statistics

For all statistical analyses, confirm that the following items are present in the figure legend, table legend, main text, or Methods section.

| n/a | Confirmed | |
|---|---|---|
| ☐ | ☒ | The exact sample size (*n*) for each experimental group/condition, given as a discrete number and unit of measurement |
| ☐ | ☒ | A statement on whether measurements were taken from distinct samples or whether the same sample was measured repeatedly |
| ☐ | ☒ | The statistical test(s) used AND whether they are one- or two-sided *Only common tests should be described solely by name; describe more complex techniques in the Methods section.* |
| ☐ | ☒ | A description of all covariates tested |
| ☒ | ☐ | A description of any assumptions or corrections, such as tests of normality and adjustment for multiple comparisons |
| ☐ | ☒ | A full description of the statistical parameters including central tendency (e.g. means) or other basic estimates (e.g. regression coefficient) AND variation (e.g. standard deviation) or associated estimates of uncertainty (e.g. confidence intervals) |
| ☐ | ☒ | For null hypothesis testing, the test statistic (e.g. *F*, *t*, *r*) with confidence intervals, effect sizes, degrees of freedom and *P* value noted *Give P values as exact values whenever suitable.* |
| ☒ | ☐ | For Bayesian analysis, information on the choice of priors and Markov chain Monte Carlo settings |
| ☒ | ☐ | For hierarchical and complex designs, identification of the appropriate level for tests and full reporting of outcomes |
| ☐ | ☒ | Estimates of effect sizes (e.g. Cohen's *d*, Pearson's *r*), indicating how they were calculated |

*Our web collection on statistics for biologists contains articles on many of the points above.*

## Software and code

Policy information about availability of computer code

| Data collection | Confocal mages were acquired below pixel saturation using Confocal Laser Scanning Microscope Leica TCS SP8 (Leica microsystem) or Nikon Inverted confocal spinning-disk microscope or a Plan-Apochromat 63x or 100×/1.4 oil objective on a Zeiss LSM800 or LSM880 confocal system equipped with an AiryScan module and controlled by the Zen blue software; FACS analysis of Sytox Green uptake was performed using BD FACS Celesta™ Cell Analyzer (BD Bioscience); for FLIM-FRET analysis, samples were excited using a pulsed laser (femtosecond Ti:Sa laser, Chameleon VISION 2 from Coherent, set at 900 nm). Photons were temporally collected using a single-photon sensitive detector (PMA-Hybrid 40; Picoquant) combined with a single photon counting module (TimeHarp 260; Picoquant). |
|---|---|
| Data analysis | Image J-based Fiji, Fiji co-localization plug-in "Coloc2", "Analyze particles" tool of Fiji, FlowJo 10.8.1 software, GraphPad Prism (GraphPad Software). FLIM data analysis was performed using SymPhoTime 64 (Picoquant). |

For manuscripts utilizing custom algorithms or software that are central to the research but not yet described in published literature, software must be made available to editors and reviewers. We strongly encourage code deposition in a community repository (e.g. GitHub). See the Nature Portfolio guidelines for submitting code & software for further information.

## Data

Policy information about availability of data

All manuscripts must include a data availability statement. This statement should provide the following information, where applicable:
- Accession codes, unique identifiers, or web links for publicly available datasets
- A description of any restrictions on data availability
- For clinical datasets or third party data, please ensure that the statement adheres to our policy

All data is available in the main text, extended data figures or the supplementary materials.

## Human research participants

Policy information about studies involving human research participants and Sex and Gender in Research.

| | |
|---|---|
| Reporting on sex and gender | No human research participants invovled |
| Population characteristics | No human research participants invovled |
| Recruitment | No human research participants invovled |
| Ethics oversight | No human research participants invovled |

Note that full information on the approval of the study protocol must also be provided in the manuscript.

# Field-specific reporting

Please select the one below that is the best fit for your research. If you are not sure, read the appropriate sections before making your selection.

☒ Life sciences    ☐ Behavioural & social sciences    ☐ Ecological, evolutionary & environmental sciences

For a reference copy of the document with all sections, see nature.com/documents/nr-reporting-summary-flat.pdf

# Life sciences study design

All studies must disclose on these points even when the disclosure is negative.

| | |
|---|---|
| Sample size | For LPS-induced mortality, the sample size was calculated according to the section "Appendix A: Sample size determination" of "Guideline for the care and use of mammals in neuroscience and beharioral research 2003" (PMID: 20669478). In Arfrp1-floxed mice, mortality after LPS injection was estimated at 20%; the expected mortality in myeloid-specific Arfrp1 knockout mice was to be around 70-80%. Based on this, the calculated sample size in each group is 16-18 mice to detect the differences between groups with a statistical power statistical power of 90% and a significance level of 5%. For the rest, no calculation of sample size was performed. A minimum of 3 biologically replicates were measured and analyzed. All the experiments were repeated for at least 3 times with similar results. |
| Data exclusions | No data exclusions was performed. |
| Replication | All the experiments have been repeated for at least 3 times. All the attempts of replication were successful. |
| Randomization | For in vitro experiment, cells from the same pool were randomly split into separate wells and subjected to indicated treatments. For in vivo experiments, mice with the same gender were randomly grouped for treatments. Each group contained similar number of males and females to avoid gender-based effects. The results from both males and females in the same group were pooled and analyzed. |
| Blinding | No subjective decision making is required for the experiments shown. Investigators were not blinded. There was no need for blinding. |

# Reporting for specific materials, systems and methods

We require information from authors about some types of materials, experimental systems and methods used in many studies. Here, indicate whether each material, system or method listed is relevant to your study. If you are not sure if a list item applies to your research, read the appropriate section before selecting a response.

## Materials & experimental systems

| n/a | Involved in the study |
|-----|------------------------|
| ☐ | ☒ Antibodies |
| ☐ | ☒ Eukaryotic cell lines |
| ☒ | ☐ Palaeontology and archaeology |
| ☐ | ☒ Animals and other organisms |
| ☒ | ☐ Clinical data |
| ☒ | ☐ Dual use research of concern |

## Methods

| n/a | Involved in the study |
|-----|------------------------|
| ☒ | ☐ ChIP-seq |
| ☐ | ☒ Flow cytometry |
| ☒ | ☐ MRI-based neuroimaging |

# Antibodies

| | |
|---|---|
| Antibodies used | Antibodies used in this study for immunofluorescence analysis are: goat anti-human IL-1β antibody (AF-201-NA, R&D Systems; 1/1000); mouse anti-human Caspase1 antibody (06-503, Merck Millipore; 1/1000); rabbit anti-mouse IL-1β antibody (5129-100, BioVision; 1/1000); mouse anti-mouse Caspase1 p20 (AG-20B-0042-C100, AdipoGen; 1/1000); rabbit anti-ASC antibody (sc-22514, Santa Cruz Biotechnology; 1/1000), mouse anti-NLRP3 antibody (G-20B-0014-C100, AdipoGen; 1/3000); rabbit anti-GAPDH (G9545, Sigma-Aldrich; 1/5000); mouse anti-Tubulin (T9026, Sigma-Aldrich; 1/5000); rabbit anti-ARFRP1 (PA5-50606, ThermoFisher Scientific; 1/2000) and mouse anti-Flag antibody (F1804, Sigma-Aldrich; 1/5000); rabbit anti-OSBP (11096-1-AP, Proteintech; 1/2000); mouse anti-PI4KII alpha (sc-390026, Santa Cruz Biotechnology; 1/1000); rabbit anti-PI4KIIβ (A17719, ABclonal; 1/1000) and mouse anti-PI4KIIIβ (611816, BD Bioscience; 1/1000). Rabbit polyclonal antibodies against VAPA (; 1/6000) and VAPB (1/4000) were generated as described in Venditti et al, 2019 (PMID: 30659099). HRP-conjugated rabbit anti-goat IgG (31402, ThermoFisher Scientific, 1/10000), HRP-conjugated goat anti-rabbit IgG (111-035-144, Jackson ImmunoResearch, 1/10000); goat anti-mouse IgG (31430, ThermoFisher Scientific, 1/10000). Antibodies used in this study for immunoblotting analysis are: Rabbit anti-TGN46 (13573-1-AP, ProteinTech, 1/300); Sheep anti-human TGN46 (AHP500G, Bio-rad, 1/50); Sheep anti-TGN38 (AHP499G, Bio-rad, 1/50); Rabbit anti-GCC2 (HPA035849, Sigma-Aldrich, 1/200); Rabbit anti-GCC1 (HPA021323, Sigma-Aldrich, 1/100); Mouse anti-human p230 (611280, BD Bioscience, 1/100); Mouse anti-Golgin97 (A-21270, ThermoFisher Scientific, 1/100); Mouse anti-EEA1 (610456, BD Bioscience, 1/100); Rabbit anti-EEA1 (3288S, Cell Signaling Technology, 1/100); Mouse anti-PI4P (Z-P004, Echelon Biosciences, 1/100); Rabbit anti-Golgin97 (Home-made, PMID30659099, 1/100); Alexa Fluor488 goat anti-mouse IgG (A11029, ThermoFisher Scientific, 1/1000); Alexa Fluor594 goat anti-mouse IgG (A11005, ThermoFisher Scientific, 1/1000); Alexa Fluor488 goat anti-mouse IgM (A21042, ThermoFisher Scientific, 1/1000); Alexa Fluor594 goat anti-mouse IgM (A11029, ThermoFisher Scientific, 1/1000); Alexa Fluor488 goat anti-rabbit IgG (A11034, ThermoFisher Scientific, 1/1000); Alexa Fluor594 goat anti-rabbit IgG (A11037, ThermoFisher Scientific, 1/1000); Alexa Fluor647 goat anti-rabbit IgG (A21246, ThermoFisher Scientific, 1/1000) and Alexa Fluor488 donkey anti-sheep IgG (A11015, ThermoFisher Scientific, 1/1000). |
| Validation | Goat anti-human IL-1β antibody (AF-201-NA, R&D Systems); mouse anti-human Caspase-1 antibody (06-503, Merck Millipore); rabbit anti-mouse IL-1β antibody (5129-100, BioVision); mouse anti-mouse Caspase-1 p20 (AG-20B-0042-C100, AdipoGen); rabbit anti-ASC antibody (sc-22514, Santa Cruz Biotechnology), mouse anti-NLRP3 antibody (G-20B-0014-C100, AdipoGen) were widely used in inflammasome-related studies, including our previous study (PMID: 28716882);  rabbit anti-VAPA (home-made); rabbit anti-VAPB (home-made); rabbit anti-OSBP (11096-1-AP, Proteintech); mouse anti-PI4KII alpha(sc-390026, Santa Cruz Biotechnology), rabbit anti-PI4KII beta (A17719, ABclonal) and mouse anti-PI4KIIIβ antibody (611816, BD Bioscience) were validated in this study in Fig. 3c, 3d, Extended Data Fig. 3b, 3e; rabbit anti-ARFRP1 (PA5-50606, ThermoFisher Scientific); rabbit anti-GCC2 antibody (HPA035849, Sigma-Aldrich); rabbit anti-GCC1 antibody (HPA021323, Sigma-Aldrich); mouse anti-human p230 (611280, BD Bioscience) and mouse anti-Golgin97 (A-21270, ThermoFisher Scientific) were validated in previous study (PMID: 31575603); mouse anti-EEA1 (610456, BD Bioscience), rabbit anti-EEA1 (3288S, Cell Signaling Technology), sheep anti-human TGN46 (AHP500GT, Bio-rad) and rabbit anti-Golgin97 were validated by previous studies (including PMID: 30659099); mouse anti-PI4P (Z-P004, Echelon Biosciences) was validated by previous studies (including PMID: 30659099 and PMID: 19508231). |

# Eukaryotic cell lines

Policy information about cell lines and Sex and Gender in Research

| | |
|---|---|
| Cell line source(s) | THP-1 cells was from ATCC; HEK293t cells (ATCC, CRL-3216) was from DKFZ Heidelberg; Immortalized bone marrow-derived macrophages (iBMDMs) cell line was obtained from Dr. Eicke Latz (University of Bonn, Germany). mApple-tagged RAB5 stably-expressing HeLa cell line was provided by Dr. Anne Spang (BIOZENTRUM, University of Basel, Switzerland). HeLa WT and HeLa VAP dKO cell lines10 were a generous gift from Dr. Pietro De Camilli (Yale University School of Medicine, New Haven, CT). |
| Authentication | THP-1 cells and HEK293t cells have been authenticated using Short Tandem Repeat (STR) performed by LGC Standards, UK. HeLa cells and iBMDMs were not authenticated. |
| Mycoplasma contamination | All the cells used in this study were tested Mycoplasma-negative. |
| Commonly misidentified lines (See ICLAC register) | No misidentified cell lines were used in this study. |

# Animals and other research organisms

Policy information about <u>studies involving animals</u>; <u>ARRIVE guidelines</u> recommended for reporting animal research, and <u>Sex and Gender in Research</u>

| | |
|---|---|
| Laboratory animals | Mice were housed under specific pathogen-free conditions with controlled temperature (19-23°C) and humidity (40-60%) on a 12-h light/dark cycle with unrestricted access to water and standard laboratory chow. Arfrp1-floxed and myeloid-specific Arfrp1 knockout mice on C57BL/6J background at age 6-8 weeks-old were used. |
| Wild animals | No wild animals were used in this study. |
| Reporting on sex | Both males and females were used in this study. |
| Field-collected samples | No field-collected samples were used in this study. |
| Ethics oversight | Maintenance and animal experimentation were in accordance with the local ethical committee (Com'Eth) in compliance with the European legislation on care and use of laboratory animals (La cellule AFiS (Animaux utilisés à des Fins Scientifiques): APAFIS#30865-2021040115389039 v3) |

Note that full information on the approval of the study protocol must also be provided in the manuscript.

# Flow Cytometry

## Plots

Confirm that:

☒ The axis labels state the marker and fluorochrome used (e.g. CD4-FITC).

☒ The axis scales are clearly visible. Include numbers along axes only for bottom left plot of group (a 'group' is an analysis of identical markers).

☒ All plots are contour plots with outliers or pseudocolor plots.

☒ A numerical value for number of cells or percentage (with statistics) is provided.

## Methodology

| | |
|---|---|
| Sample preparation | After treatment, cells were detached with 5 mM EDTA and incubated with SYTOX™ Green (S7020, ThermoFisher Scientific) at 1/8000 dilution. |
| Instrument | BD FACS Celesta™ Cell Analyzer (BD Bioscience) |
| Software | FlowJo 10.8.1 Software |
| Cell population abundance | 80~90% of VAP dKO, OSBP KO, ARFRP1 or SYS1 KO THP-1 cells became SYTOX Green-positive when treated with 1 μg/ml LPS or Pam3CSK4; while WT cells showed less than 10% of SYTOX Green-positive cells. |
| Gating strategy | Forward and side scatter and SYTOX Green signal-based gating |

☒ Tick this box to confirm that a figure exemplifying the gating strategy is provided in the Supplementary Information.

