## [Peer Review File · Nature Immunology]

Peer Review Information

Journal: Nature Immunology

Manuscript Title: Distinct changes in endosomal composition promote NLRP3 inflammasome activation

Corresponding author name(s): Zhirong Zhang, Maria Antonietta De Matteis, Romeo Ricci

Editorial Notes:

Transferred manuscripts This manuscript has been previously reviewed at another journal that is not operating a transparent peer review scheme. This document only contains reviewer comments, rebuttal and decision letters for versions considered at Nature **Immunology**.

Redactions – transferred manuscripts (mention of previous referee reports from elsewhere) This manuscript has been previously reviewed at another journal. This document only contains reviewer comments, rebuttal and decision letters for versions considered at Nature **Immunology**. Mentions of prior referee reports have been redacted

Reviewer Comments & Decisions:

Decision Letter, initial version:

Subject: Decision on Nature Immunology submission NI-A34366-T

Message: 31st Aug 2022

Dear Professor Ricci,

Your Article, "Distinct changes in endosomal composition promote NLRP3 inflammasome activation" has now been seen by 2 referees. You will see from their comments below that while they find your work of interest, some important points are raised. We are very interested in the possibility of publishing your study in Nature Immunology, but would like to consider your response to these concerns in the form of a revised manuscript before we make a final decision on publication.

In line with the comments of Reviewer #1 please tone down statements on NLRP3-PI(4)P interactions.

We therefore invite you to revise your manuscript taking into account all reviewer and editor comments. Please highlight all changes in the manuscript text file in Microsoft Word format.

* If you have not done so already please begin to revise your manuscript so that it conforms to our Article format instructions at <http://www.nature.com/ni/authors/index.html>. Refer also to any guidelines provided in this letter.

* Please include a revised version of any required reporting checklist. It will be available to referees to aid in their evaluation of the manuscript goes back for peer review. They are available here:

Reporting summary:

[REDACTED]

Note: This URL links to your confidential home page and associated information about manuscripts you may have submitted, or that you are reviewing for us.

If you wish to forward this email to co-authors, please delete the link to your homepage.

We hope to receive your revised manuscript within two weeks. If you cannot send it within this time, please let us know. We will be happy to consider your revision so long as nothing similar has been accepted for publication at Nature Immunology or published elsewhere.

Nature Immunology is committed to improving transparency in authorship. As part of our efforts in this direction, we are now requesting that all authors identified as 'corresponding author' on published papers create and link their Open Researcher and Contributor Identifier (ORCID) with their account on the Manuscript Tracking System (MTS), prior to acceptance. ORCID helps the scientific community achieve unambiguous attribution of all scholarly contributions. You can create and link your ORCID from the home page of the MTS by clicking on 'Modify my Springer Nature account'. For more information please visit www.springernature.com/orcid.

Sincerely,

Stephanie Houston
Editor
Nature Immunology

Reviewers' Comments:

Reviewer #1:

Remarks to the Author:

In this study, the authors perform a detailed cell biological analysis to identify the subcellular sites of NLRP3 activation. This is an important topic for the field, as studies in recent years have identified several organelles as the site of NLRP3 inflammasome activation. In particular, mitochondria, Trans Golgi Network (TGN) and the microtubule organizing center (MTOC) have been reported to function in this manner. This present study focused on the previously reported interactions between NLRP3 and membranes containing the phosphoinositide PI(4)P. These membranes were annotated in a 2018 Nature paper by Chen and colleagues as TGN.

The authors here provide compelling evidence that the TGN is not the site of NLRP3 activation, and that the previously reported organelles containing NLRP3 are in fact, endosomes that contain PI(4)P. Detailed cell biological dissection of this process leaves little room for alternative interpretation and ultimately forces a re-definition of the sites of NLRP3 signaling. TGN vesicles are not the sites, as suggested by prior work. This finding is not trivial and should help focus future research in the field on endosomes as the sites of inflammasome signaling. The authors should highlight other work in this area, where endosomes have emerged as sites of other PRR signaling pathways. For example

pathways mediated by TLRs originated from endosomes, including TLR4, whose ligand is extensively used as a priming agent in this study.

I commend the authors of the detailed analysis in this area, which is not summarized in this critique but is nevertheless impressive.

The only experimental concern I have is the definitive nature of statements made by the authors on NLRP3-PI(4)P interactions. It is clear from this work that NLRP3 colocalizes with PI(4)P and that this phosphoinositide is important for controlling NLRP3 signaling. But there is no evidence presented in this study that NLRP3 interacts with PI(4)P, yet the authors state this interaction multiple times. In fact, even the original study on this topic (indicated above) did not show NLRP3 interaction with PI(4)P. The authors of that prior study showed a small peptide derived from NLRP3 interacts with PI(4)P. No interaction between the full length protein and this lipid has been reported. If the authors want to make these statements, I would suggest that they provide experimental evidence supporting the interaction. Otherwise, the authors are encouraged to highlight this point in their narrative, as it is an important issue for the field.

Reviewer #2:

Remarks to the Author:

The manuscript by Ricci and colleagues has confirmed data from a previous report that PI4P accumulation on membranes is essential for NLRP3 activation. The significant new finding reported in this manuscript is that the PI4P accumulation does not occur on disrupted Golgi membranes but rather on endosomes. The authors provide convincing evidence that NLRP3 activators disturb ER-endosome membrane contact sites and increase PI4P on endosomes.

The authors [REDACTED].

This manuscript corrects a quasi 'dogma' in the field that Golgi disruption is the fundamental mechanism of NLRP3 activation. I support the publication of this carefully done study in NI.

Author Rebuttal to Initial comments

Response to Referee 1's comments:

We highly appreciate the very positive assessment of our work in the context of the existing literature by referee 1.

We fully agree with her/his remaining concern that binding of full length NLRP3 to PI4P has not been evidenced experimentally so far. We do not have additional results in this context at the moment and therefore, as suggested, tuned-down all related statements and now mention it in the discussion section:

"We have also demonstrated that accumulation of PI4P on endosomes is critical for NLRP3 activation. However, whether full length NLRP3 directly binds PI4P on endosomes is still to be determined, as forced binding of NLRP3 with

PI4P was not sufficient to induce inflammasome assembly. Of note, a very recent study demonstrated that the inactive NLRP3 decamer was capable to bind negatively charged lipids including PI4P in vitro."

We also acknowledged the idea to discuss more broadly the endosomal role of pattern recognition receptor-mediated innate immunity. We added the following text to the discussion section:

"Endosomes have emerged as signaling hubs involved in many cellular processes including in basic innate immune responses. For instance, Toll-like receptors (TLRs), such as TLR3, 4, 7, 8, 9 and 13, are located on endosomal membranes. Endosomal location of TLR4 is critical for TRIF-mediated signaling. The endosomal location of other TLRs is critical for their functions in detecting cognate microbial ligands. In this study, we discovered that distinct changes in the composition of endosomes is a mark for activation of innate immunity mediated by NLRP3. Our findings thus define a so far unrecognized role of the endosomal compartment in Pattern-Recognition Receptor-induced innate immune responses."

Response to Referee 2's comments:

We are extremely happy that referee 2 recommends publication of our work in Nature Immunology without any further changes.

Decision Letter, first revision:

Subject: Your manuscript, NI-A34366A

Message: Our ref: NI-A34366A

15th Sep 2022

Dear Dr. Ricci,

Thank you for your patience as we've prepared the guidelines for final submission of your Nature Immunology manuscript, "Distinct changes in endosomal composition promote NLRP3 inflammasome activation" (NI-A34366A). Please carefully follow the step-by-step instructions provided in the attached file, and add a response in each row of the table to indicate the changes that you have made. Please also check and comment on any additional marked-up edits we have proposed within the text. Ensuring that each point is addressed will help to ensure that your revised manuscript can be swiftly handed over to our production team.

We would like to start working on your revised paper, with all of the requested files and forms, as soon as possible (preferably within two weeks). Please get in contact with us if

you anticipate delays.

When you upload your final materials, please include a point-by-point response to any remaining reviewer comments and please make sure to upload your checklist.

If you have not done so already, please alert us to any related manuscripts from your group that are under consideration or in press at other journals, or are being written up for submission to other journals (see: <https://www.nature.com/nature-portfolio/editorial-policies/plagiarism#policy-on-duplicate-publication> for details).

In recognition of the time and expertise our reviewers provide to Nature Immunology's editorial process, we would like to formally acknowledge their contribution to the external peer review of your manuscript entitled "Distinct changes in endosomal composition promote NLRP3 inflammasome activation". For those reviewers who give their assent, we will be publishing their names alongside the published article.

Nature Immunology offers a Transparent Peer Review option for new original research manuscripts submitted after December 1st, 2019. As part of this initiative, we encourage our authors to support increased transparency into the peer review process by agreeing to have the reviewer comments, author rebuttal letters, and editorial decision letters published as a Supplementary item. When you submit your final files please clearly state in your cover letter whether or not you would like to participate in this initiative. Please note that failure to state your preference will result in delays in accepting your manuscript for publication.

Cover suggestions

As you prepare your final files we encourage you to consider whether you have any images or illustrations that may be appropriate for use on the cover of Nature Immunology.

Nature Immunology has now transitioned to a unified Rights Collection system which will allow our Author Services team to quickly and easily collect the rights and permissions required to publish your work. Approximately 10 days after your paper is formally accepted, you will receive an email in providing you with a link to complete the grant of

rights. If your paper is eligible for Open Access, our Author Services team will also be in touch regarding any additional information that may be required to arrange payment for your article.

Please note that *Nature Immunology* is a Transformative Journal (TJ). Authors may publish their research with us through the traditional subscription access route or make their paper immediately open access through payment of an article-processing charge (APC). Authors will not be required to make a final decision about access to their article until it has been accepted. [Find out more about Transformative Journals](https://www.springernature.com/gp/open-research/transformative-journals).

If you have any questions about costs, Open Access requirements, or our legal forms, please contact ASJournals@springernature.com.

Please use the following link for uploading these materials: [REDACTED]

Best regards,

Elle Morris
Senior Editorial Assistant
Nature Immunology
Phone: 212 726 9207
Fax: 212 696 9752
E-mail: immunology@us.nature.com

On behalf of

Stephanie Houston
Editor
Nature Immunology

Final Decision Letter:

Subject: Decision on Nature Immunology submission NI-A34366B

Message: In reply please quote: NI-A34366B

Dear Dr. Ricci,

I am delighted to accept your manuscript entitled "Distinct changes in endosomal composition promote NLRP3 inflammasome activation" for publication in an upcoming issue of Nature Immunology.

Over the next few weeks, your paper will be copyedited to ensure that it conforms to Nature Immunology style. Once your paper is typeset, you will receive an email with a link to choose the appropriate publishing options for your paper and our Author Services team will be in touch regarding any additional information that may be required.

Please note that *Nature Immunology* is a Transformative Journal (TJ). Authors may publish their research with us through the traditional subscription access route or make their paper immediately open access through payment of an article-processing charge (APC). Authors will not be required to make a final decision about access to their article until it has been accepted. [Find out more about Transformative Journals](https://www.springernature.com/gp/open-research/transformative-journals).

Authors may need to take specific actions to achieve [compliance](https://www.springernature.com/gp/open-research/funding/policy-compliance-faqs) with funder and institutional open access mandates. If your research is supported by a funder that requires immediate open access (e.g. according to [Plan S principles](https://www.springernature.com/gp/open-research/plan-s-compliance)) then you should select the gold OA route, and we will direct you to the compliant route where possible. For authors selecting the subscription publication route, the journal's standard licensing terms will need to be accepted, including [8](https://www.springernature.com/gp/open-research/policies/journal-self-archiving policies. Those licensing terms will supersede any other terms that the author or any third party may assert apply to any version of the manuscript.

Your paper will be published online soon after we receive your corrections and will appear in print in the next available issue. Content is published online weekly on Mondays and Thursdays, and the embargo is set at 16:00 London time (GMT)/11:00 am US Eastern time (EST) on the day of publication. Now is the time to inform your Public Relations or Press Office about your paper, as they might be interested in promoting its publication. This will allow them time to prepare an accurate and satisfactory press release. Include your manuscript tracking number (NI-A34366B) and the name of the journal, which they will need when they contact our office.

About one week before your paper is published online, we shall be distributing a press release to news organizations worldwide, which may very well include details of your work. We are happy for your institution or funding agency to prepare its own press release, but it must mention the embargo date and Nature Immunology. Our Press Office will contact you closer to the time of publication, but if you or your Press Office have any enquiries in the meantime, please contact press@nature.com.

Also, if you have any spectacular or outstanding figures or graphics associated with your manuscript - though not necessarily included with your submission - we'd be delighted to consider them as candidates for our cover. Simply send an electronic version (accompanied by a hard copy) to us with a possible cover caption enclosed.

If you have not already done so, we strongly recommend that you upload the step-by-step protocols used in this manuscript to the Protocol Exchange. Protocol Exchange is an open online resource that allows researchers to share their detailed experimental know-how. All uploaded protocols are made freely available, assigned DOIs for ease of citation and fully searchable through nature.com. Protocols can be linked to any publications in which they are used and will be linked to from your article. You can also establish a dedicated page to collect all your lab Protocols. By uploading your Protocols to Protocol Exchange, you are enabling researchers to more readily reproduce or adapt the methodology you use, as well as increasing the visibility of your protocols and papers. Upload your Protocols at www.nature.com/protocolexchange/. Further information can be found at

www.nature.com/protocolexchange/about .

Please note that we encourage the authors to self-archive their manuscript (the accepted version before copy editing) in their institutional repository, and in their funders' archives, six months after publication. Nature Portfolio recognizes the efforts of funding bodies to increase access of the research they fund, and strongly encourages authors to participate in such efforts. For information about our editorial policy, including license agreement and author copyright, please visit www.nature.com/ni/about/ed_policies/index.html

Sincerely,

Stephanie Houston
Editor
Nature Immunology